# HSP47 levels determine the degree of body adiposity

Jihoon Shin [1,2,3] ✉, Shinichiro Toyoda[1], Yosuke Okuno[1], Reiko Hayashi[1], Shigeki Nishitani[1], Toshiharu Onodera[1,4], Haruyo Sakamoto[1], Shinya Ito[5], Sachiko Kobayashi [1], Hirofumi Nagao [1], Shunbun Kita [1,6], Michio Otsuki[1,7], Atsunori Fukuhara [1,6], Kazuhiro Nagata[8,9] & Iichiro Shimomura[1]

Adiposity varies among individuals with the influence of diverse physiological, pathological, environmental, hormonal, and genetic factors, but a unified molecular basis remains elusive. Here, we identify HSP47, a collagen-specific chaperone, as a key determinant of body adiposity. HSP47 expression is abundant in adipose tissue; increased with feeding, overeating, and obesity; decreased with fasting, exercise, calorie restriction, bariatric surgery, and cachexia; and correlated with fat mass, BMI, waist, and hip circumferences. Insulin and glucocorticoids, respectively, up- and down-regulate HSP47 expression. In humans, the increase of *HSP47* gene expression by its intron or synonymous variants is associated with higher body adiposity traits. In mice, the adipose-specific knockout or pharmacological inhibition of HSP47 leads to lower body adiposity compared to the control. Mechanistically, HSP47 promotes collagen dynamics in the folding, secretion, and interaction with integrin, which activates FAK signaling and preserves PPARγ protein from proteasomal degradation, partly related to MDM2. The study highlights the significance of HSP47 in determining the amount of body fat individually and under various circumstances.

Adipose tissue, composed of adipocytes, is a specialized organ to store energy as fat, thus commonly referred to as fat tissue[1]. It is distributed throughout the body, including subcutaneous fat beneath the skin, visceral fat around internal organs, and brown fat in the back, neck, and shoulder area[2]. Fat tissue can dynamically change its size and mass in response to nutritional and hormonal states. During eating or overeating conditions, increased blood insulin levels stimulate glucose uptake and lipid accumulation in fat cells, promoting the expansion of fat tissue[3]. Conversely, fasting or starving conditions induce glucocorticoid hormone, which triggers the lipolysis of stored fat, leading to

a reduction in fat tissue[4]. The peroxisome proliferator-activated receptor-γ (PPARγ) is a master regulator of fat cells[5]. Defects in PPARγ result in the loss of fat tissue[6–9], while its activation by agonizts, such as pioglitazone, promotes lipid storage and the expansion of fat tissue[10,11]. The unique properties of fat tissue and its regulation by various nutritional, hormonal, and molecular factors have been well established, yet the intricate connections between these elements is not fully understood.

Adiposity, referring to the amount or percentage of body fat, varies greatly between individuals, ranging from as low as 5% in very

[1]Department of Metabolic Medicine, Graduate School of Medicine, Osaka University, Suita, Osaka, Japan. [2]Department of Diabetes Care Medicine, Graduate School of Medicine, Osaka University, Suita, Osaka, Japan. [3]Division of Endocrinology, Diabetes and Metabolism, Beth Israel Deaconess Medical Center and Harvard Medical School, Boston, MA, USA. [4]Touchstone Diabetes Center, Department of Internal Medicine, The University of Texas Southwestern Medical Center, Dallas, USA. [5]Faculty of Life Sciences, Kyoto Sangyo University, Kyoto, Japan. [6]Department of Adipose Management, Graduate School of Medicine, Osaka University, Suita, Osaka, Japan. [7]Department of Endocrinology, Graduate School of Medical Science, Tokyo Women's Medical University, Tokyo, Japan. [8]Institute for Protein Dynamics, Kyoto Sangyo University, Kyoto, Japan. [9]JT Biohistory Research Hall, Osaka, Japan. ✉e-mail: shinjihoon0209@gmail.com

thin/lean people to over 40% in the cases of morbid obesity[12]. Additionally, each person exhibits a unique susceptibility to gain body fat under regular eating or overeating conditions[13]. These individual variations are influenced by a combination of physiological, environmental, pathological, and genetic factors. Physiologically, eating habits and patterns, which affect food calorie intake, fasting duration, and hormone levels, have a significant impact on body adiposity[14,15]. Environmentally, engaging in high physical activity or exercise can also contribute to lower body adiposity[14,15]. Pathologically, obesity is characterized by high levels of body fat tissue[15], while cachexia, a wasting disorder, leads to a significant loss of fat tissue[16]. Bariatric surgery has been shown effective results in reducing body adiposity in patients with obesity[17]. Genetic factors, such as single nucleotide polymorphisms (SNPs), also play a role in determining body adiposity traits, such as fat mass, BMI, waist circumference, and hip circumferences[18]. Despite the comprehensive contribution of physiological, pathological, environmental, and genetic backgrounds to body adiposity, a unified molecular basis for its regulation remains elusive.

In this study, through a series of in silico, in vivo, and in vitro experiments, we identify HSP47, a collagen-specific molecular chaperone[19,20], as a significant determinant of body adiposity. Our transcriptomic and genetic analyses indicate that HSP47 is prominently expressed in adipose tissue and adipocytes, with its levels tightly linked to the degree of body adiposity in humans. Adipose-specific and pharmacological ablation models in mice provide compelling evidence of the causal impact of HSP47 on body fat deposition. Biochemical assays reveal that HSP47-meidated enhancement of collagen protein dynamics, such as its folding, secretion, and binding in the extracellular matrix, induce focal adhesion signaling, stabilize PPARγ protein, and ultimately lead to the expansion of fat tissue. This research not only deepens our understanding of the individual and contextual factors that determine body adiposity but also provides a robust scientific rationale for these mechanisms.

## Results

### Collagen matrix, focal adhesion, and PPARγ characterize fat tissue

Each organ/tissue has its unique characteristics, which are defined by their specific gene expression patterns[21]. To understand the characteristics of fat tissue in transcription levels, we compared the gene expression patterns of adipose tissue with those of other 52 different organs and tissues. We scored the adipose enrichment of each transcript and plotted it against the gene expression level. As expected, ADIPOQ (Adiponectin), LEP (Leptin), and FABP4 (aP2) genes were found to be highly enriched and expressed in adipose tissue (Fig. 1a). We performed gene ontology (GO) analysis on the top 500 genes and found that components of the collagen-containing ECM and focal adhesion were enriched in fat tissue (Fig. 1b). Pathway analysis further revealed that these genes are involved in the PPAR signaling pathway, ECM-receptor interaction, and focal adhesion (Fig. 1c). Focal adhesion is subcellular structures composed of multiple proteins, including Integrins and focal adhesion kinase (FAK), that serves as a link between the extracellular environment and intracellular compartments[22,23]. Therefore, the interactions between extracellular collagen matrix, subcellular focal adhesion, and intracellular PPARγ may play a crucial role in shaping the characteristics of adipose tissue.

### HSP47 levels associate with the amount of body fat

Individual variations in body adiposity are associated with their physiological, pathological, environmental, and genetic backgrounds. To explore potential determinants of body adiposity, we conducted a screening using public databases (e.g., GTEx[21], GEO[24], GWAS catalog[25]) and compiled gene lists associated with high adiposity conditions in human adipose tissues and cells (Fig. 1d): Genes that are (1) enriched in human adipose tissue, (2) enhanced during feeding in human adipose

tissue, (3) induced by overeating (8 weeks of high-fat diet) in human subcutaneous adipose tissue, (4) upregulated by obesity in human adipocytes, and (5) associated with various adiposity traits in human genome-wide association study (GWAS), such as waist circumferences, hip circumferences, waist/hip ratio, obesity, and BMI. Through this comparative analysis, we identified HSP47, a collagen-specific chaperone encoded by SERPINH1 gene[19,20], as a potential determinant of body adiposity.

Detailed expression profiles revealed that HSP47 gene and protein are highly expressed in adipose tissues and adipocytes compared with other tissues, organs, and cells in humans (Fig. 1e, f). Similar expression patterns were observed in mice (Supplementary Fig. 1a-c). The gene expression of HSP47 increased in human adipose tissues with feeding (Fig. 1g) and overeating (Fig. 1h). Individuals with obesity exhibited higher gene expression of HSP47 in adipocytes (Fig. 1i). There were no significant sexual differences of HSP47 expression in subcutaneous and visceral adipose tissues (Supplementary Fig. 2a); both male and female subjects displayed equivalent induction of HSP47 in obesity (Supplementary Fig. 2b). We also analyzed adipose tissues from young adult (22- to 36-year-old) monozygotic twin-pairs with discordant BMI (inter-pair difference $\Delta$BMI > 3 kg/m$^2$), which showed significant differences in various body adiposity traits such as body weight, fat mass, and volume[26]. The gene expression of HSP47 was significantly higher in the adipose tissue of high-adiposity monozygotic twin pairs compared to that of low-adiposity twin pairs (Fig. 1j), indicating that HSP47 gene expression associates with body adiposity independent of genetic backgrounds. In contrast, HSP47 gene expression was significantly lower in adipose tissues under low-adiposity conditions, such as exercise training (Fig. 1k), calorie restriction (Fig. 1l), bariatric surgery (Fig. 1m), and cachexia (Fig. 1n) compared to the controls.

Similar results were observed in rodents, including mice and rats (Supplementary Fig. 2c-n). The gene expressions of Hsp47 were increased by feeding and overeating conditions in adipose tissues and adipocytes (Supplementary Fig. 2c, d). We also analyzed the transcriptome of adipose tissues from genetically identical C57BL/6 J inbred male mice that exhibited high- or low level of weight gain and fat mass after 4 weeks on a high fat diet[27]. The gene expression of Hsp47 was significantly higher in the adipose tissue of high-weight/fat gainer mice compared to that of low-weight/fat gainer mice, suggesting a non-genetic association of the Hsp47 gene with body adiposity in mice. (Supplementary Fig. 2e). In contrast, the gene expression of Hsp47 in adipose tissues was decreased by fasting, calorie restriction, exercise, and cachexia (Supplementary Fig. 2c, f-h). In rat models, the gene expression of Hsp47 increased with feeding and obese conditions, but decreased with starvation, calorie restriction, and bariatric surgery (Supplementary Fig. 2i, l-n). These results suggested that HSP47 gene expression is associated with various adiposity conditions in both human and rodent species.

We next examined the relationship between HSP47 gene expression in adipose tissue and various body adiposity traits. To do so, we analyzed adipose tissue samples from a large, age- and gender-matched cohort previously described in the METSIM study[28]. The analysis showed a direct and significant correlations between HSP47 gene expression in adipose tissue and fat mass, body mass index (BMI), waist circumference, and hip circumference, whereas inversely correlated with lean mass (Fig. 1o-s). These findings clearly indicate that HSP47 is associated with body adiposity.

### Insulin and glucocorticoids regulate HSP47 expression

We next investigated the hormonal regulation of HSP47 expression. Based on the expression patterns of HSP47—upregulation in feeding, overeating, and obesity, while downregulation in fasting, exercise, calorie restriction, bariatric surgery, and cachexia—we hypothesized that endocrine hormones, such as insulin and glucocorticoids may be involved in regulating HSP47 expression. Indeed, in vitro and ex vivo

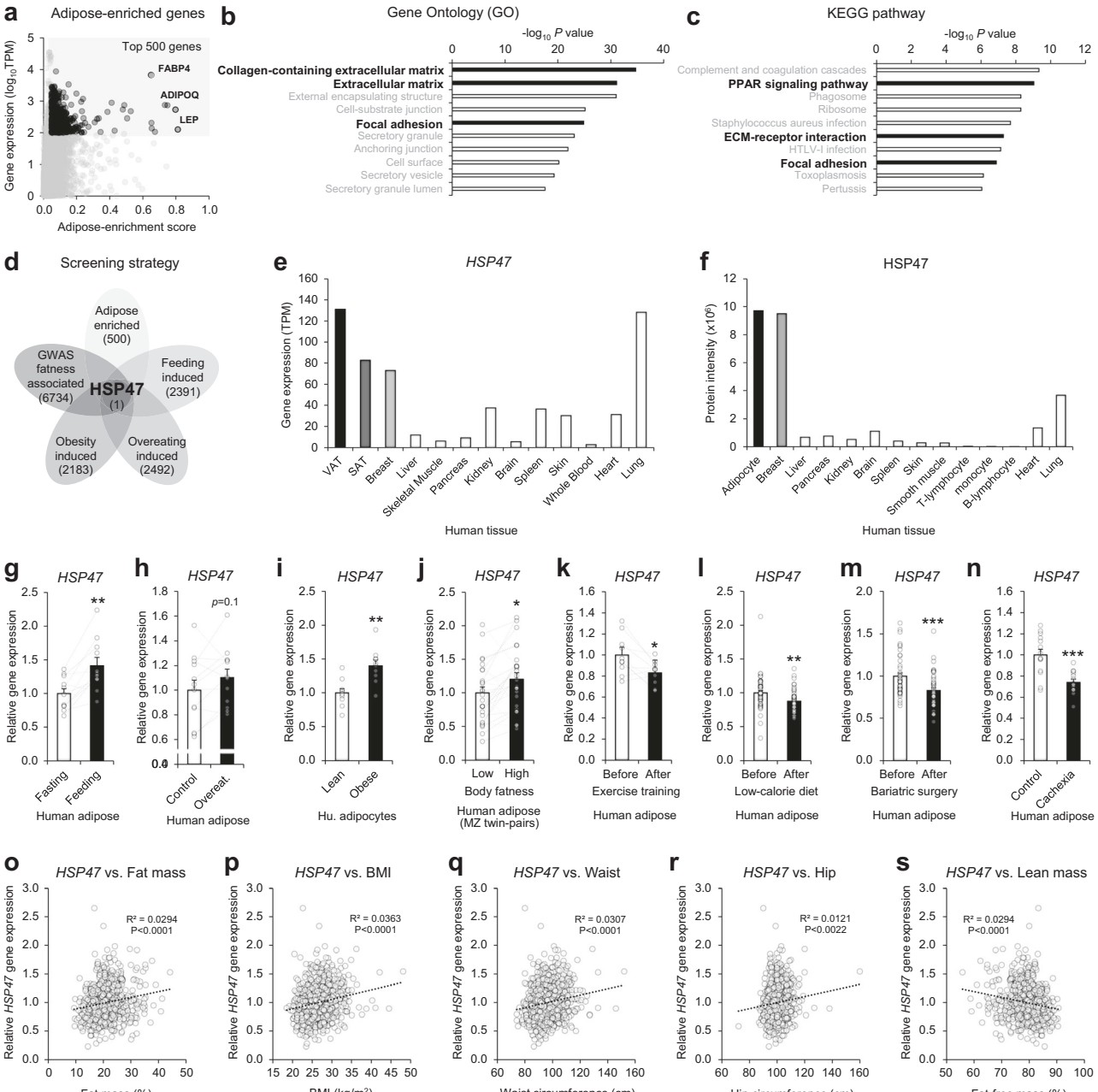

**Fig. 1 | The association of HSP47 expression with human body adiposity.**
**a** Transcript enrichment analysis of human adipose tissue compared with other various tissues and cells (GTEx). Gene ontology (**b**) and KEGG pathway (**c**) analyses of adipose-enriched genes. **d** Schematic diagram of determinant screening for body adiposity (Human adipose tissue-enriched genes GTEx: Adipose-enrichment top-ranked 500 genes from TPM > 100 in human adipose tissue; Feeding induced genes in human subcutaneous adipose tissue (GSE154612: fold change > 1.1, p value < 0.05); Overeating-induced genes in human adipose tissue (GSE28005: fold change > 1.1, p value < 0.15); Obesity-induced gene in human subcutaneous adipocytes (GSE80654: fold change > 1.1, p value < 0.15); Body adiposity-associated genes in GWAS. **e** HSP47 gene expressions in human tissues (GTEx). **f** HSP47 protein intensity extracted from Proteomics DB. **g** Human subcutaneous adipose tissue from healthy subjects under fasting and feeding conditions (GSE154612, n = 11, 2 h or 24 h after standard meal; p = 0.0013). **h** Human subcutaneous adipose tissue before and after over-eating (GSE28005, n = 12, high-fat diet for 56 days). **i** Human subcutaneous adipocytes from non-diabetic lean subjects and subjects with obesity (lean, n = 10; obese, n = 10; p = 0.0071; GDS3602). **j** Human subcutaneous adipose tissue from monozygotic twin-pairs discordant for body adiposity (GSE92405,

n = 25, intrapair difference in BMI > 3 kg/m²; p = 0.024). **k** Human subcutaneous adipose tissue from healthy young male subjects before and after 12 weeks of exercise training (GSE116801, n = 10, 60–80 min cycling/day, 5 days/week; p = 0.0114). **l** Human subcutaneous adipose tissue from overweight and obese before and after low-calorie diet interventions (GSE77962; before, n = 51; after, n = 51; 1250 kcal/d diet for 12 weeks or 500 kcal/d diet for 5 weeks; p = 0.0039). **m** Human subcutaneous adipose tissue from obese female subjects before and 1 year after bariatric surgery (GSE72158, n = 42 each, Roux-en-Y gastric bypass [RYGB]; p = 0.000521). **n** Human subcutaneous adipose tissue from gastrointestinal cancer patients with (n = 13) or without (n = 14) cachexia (GSE20571; p = 0.000276). Correlation analyses between HSP47 gene expression in human subcutaneous adipose tissue and body adiposity traits, including fat mass (**o**), BMI (**p**), waist circumference (**q**), hip circumference (**r**), lean mass (s; fat-free mass) in a cohort METSIM study (GSE70353). Data represent the mean ± SEM. *p < 0.05, **p < 0.01, and ***p < 0.001; n refers to sample size. Statistical significance was determined by Two-tailed paired (**g**, **h**, **j**, **k**, **m**), unpaired (**i**, **l**, **n**) t-test, and pearson correlation test (**o**–**s**). Source data are provided as a Source Data file.

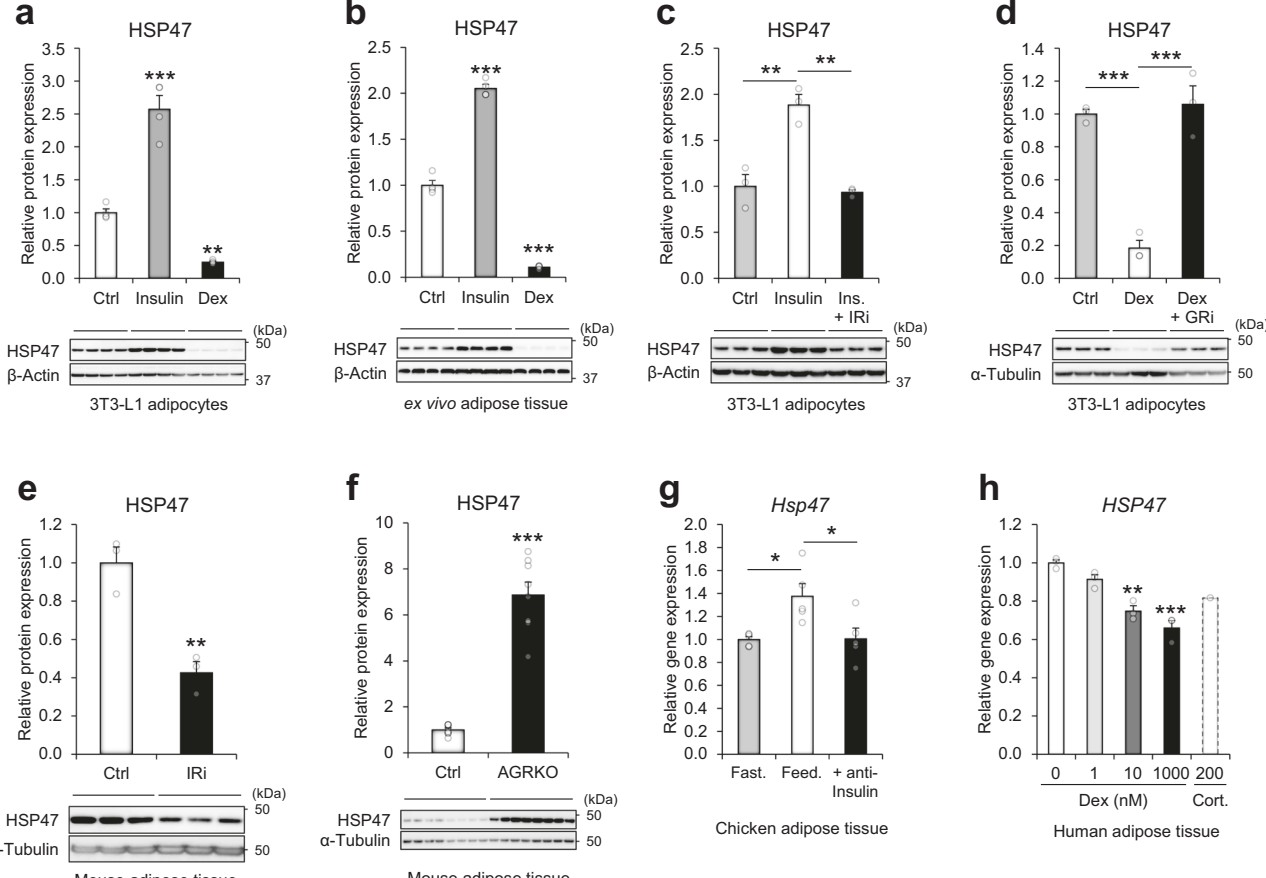

**Fig. 2 | Hormonal regulation of HSP47 by insulin and corticosteroids. a** Relative protein expression of HSP47 in 3T3-L1 adipocytes after insulin (100 nM; $p < 0.0001$) or dexamethasone (Dex;100 nM; $p < 0.0052$) treatments for 48 h ($n = 4$ each). **b**, Relative protein expression of HSP47 in ex vivo cultured mouse adipose tissue after insulin (100 nM; $p < 0.0001$) or dexamethasone (100 nM; $p < 0.0001$) treatments for 48 h ($n = 4$ each). **c** Relative protein expression of HSP47 in 3T3-L1 adipocytes after insulin (Ins; 100 nM; ctrl vs Insulin $p = 0.0013$) treatment with/without insulin receptor inhibitor (OSI-906; IRi; 0.2 μM; Insulin vs Ins.+ IRi, $p = 0.0019$) for 48 h ($n = 3$ each). **d** Relative protein expression of HSP47 in 3T3-L1 adipocytes after dexamethasone (100 nM; $p = 0.0003$) treatments with/without glucocorticoid receptor antagonist (RU486; GRi; 10 μM; Dex vs Dex+GRi, $p = 0.0005$) for 48 h ($n = 3$ each). **e** Relative protein expression of HSP47 in mouse adipose tissue after insulin receptor inhibitor (OSI-906; IRi; 100 mg/kg) treatment for 24 h ($n = 3$ each;

$p = 0.0045$). **f** Relative protein expression of HSP47 in mouse adipose tissue of control (flox/flox) or adipose-specific GR knockout (AGRKO) mice after 2 weeks of corticosterone treatment ($n = 8$; $p < 0.001$). **g** Relative gene expression of Hsp47 in chicken adipose tissue after short-term 5 h fasting (Fast.) or ad libitum feeding (Feed.) with/without insulin neutralization by porcine anti-insulin serum (GSE35581; $n = 5$ each; Fast. vs Feed. $p = 0.0194$; Feed. vs anti-Insulin, $p = 0.0215$). **h** Relative gene expression of HSP47 in ex vivo cultured human omental adipose tissue with dexamethasone (0, 1, 10, 1000 nM; $n = 3$; 10 nM, $p = 0.0014$; 1000 nM, $p = 0.0002$) or cortisol (200 nM; $n = 1$) (GSE88966). Data represent the mean ± SEM. *$p < 0.05$, **$p < 0.01$, and ***$p < 0.001$; $n$ refers to sample size. Statistical significance was determined by Tukey–Kramer test (**a**–**d**, **g**, **h**) and two-tailed unpaired $t$-test (**e** and **f**). Source data are provided as a Source Data file.

experiments demonstrated that insulin treatment increased the expression of HSP47 protein, whereas treatment with dexamethasone, a synthetic analog of cortisol, decreased it in 3T3-L1 adipocytes and adipose tissue explants (Fig. 2a, b); these effects exhibited dose dependency within the range of 0 to 100 nM (Supplementary Fig. 3a–d). The blockade of each insulin receptor (IR) or glucocorticoid receptor (GR) by specific inhibitors abrogated the insulin- or glucocorticoid-mediated HSP47 regulation in in vitro 3T3-L1 adipocytes (Fig. 2c, d). In vivo, the inhibition of insulin receptor decreased HSP47 protein expression in adipose tissue (Fig. 2e), while adipose-specific GR knockout (AGRKO) increased HSP47 protein expression (Supplementary Fig. 3e), and the effect was more pronounced under the treatment with corticosterone (Fig. 2f). These hormonal regulations of HSP47 were consistent with transcriptome data from other independent studies; the neutralization of insulin action by anti-insulin serum in vivo chicken decreased the gene expression of Hsp47 in adipose tissue, comparable to the level observed during fasting (Fig. 2g); additionally, in human adipose explants, treatment with dexamethasone or cortisol decreased HSP47 gene expression in a

dose-dependent manner (Fig. 2h). Taken together, these results demonstrated that HSP47 is regulated by insulin and glucocorticoids.

## Genetic increases of HSP47 gene associate with higher body adiposity

Individual body adiposity is significantly influenced by hereditary and genetic backgrounds. Cumulative genome-wide association studies (GWAS) have identified a large number of genetic loci associated with high or low traits of body adiposity. Several single nucleotide polymorphisms (SNPs) in HSP47 gene loci, such as intron variants (rs606452-A, rs668347-T, and rs645935-T) or a synonymous variant (rs584961-A), have been reported to be associated with higher body adiposity traits (Fig. 3a), such as waist circumference and/or hip circumference. However, the impact of these variants on the gene expression of HSP47 itself has not yet been elucidated. To assess this impact, we performed an expression quantitative trait loci (eQTL) analysis and found that body adiposity-related HSP47 gene variants (rs606452-A, rs668347-T, rs645935-T, and rs584961-A), but not other HSP47 gene variants (rs605040 and rs646474) with no reports of body

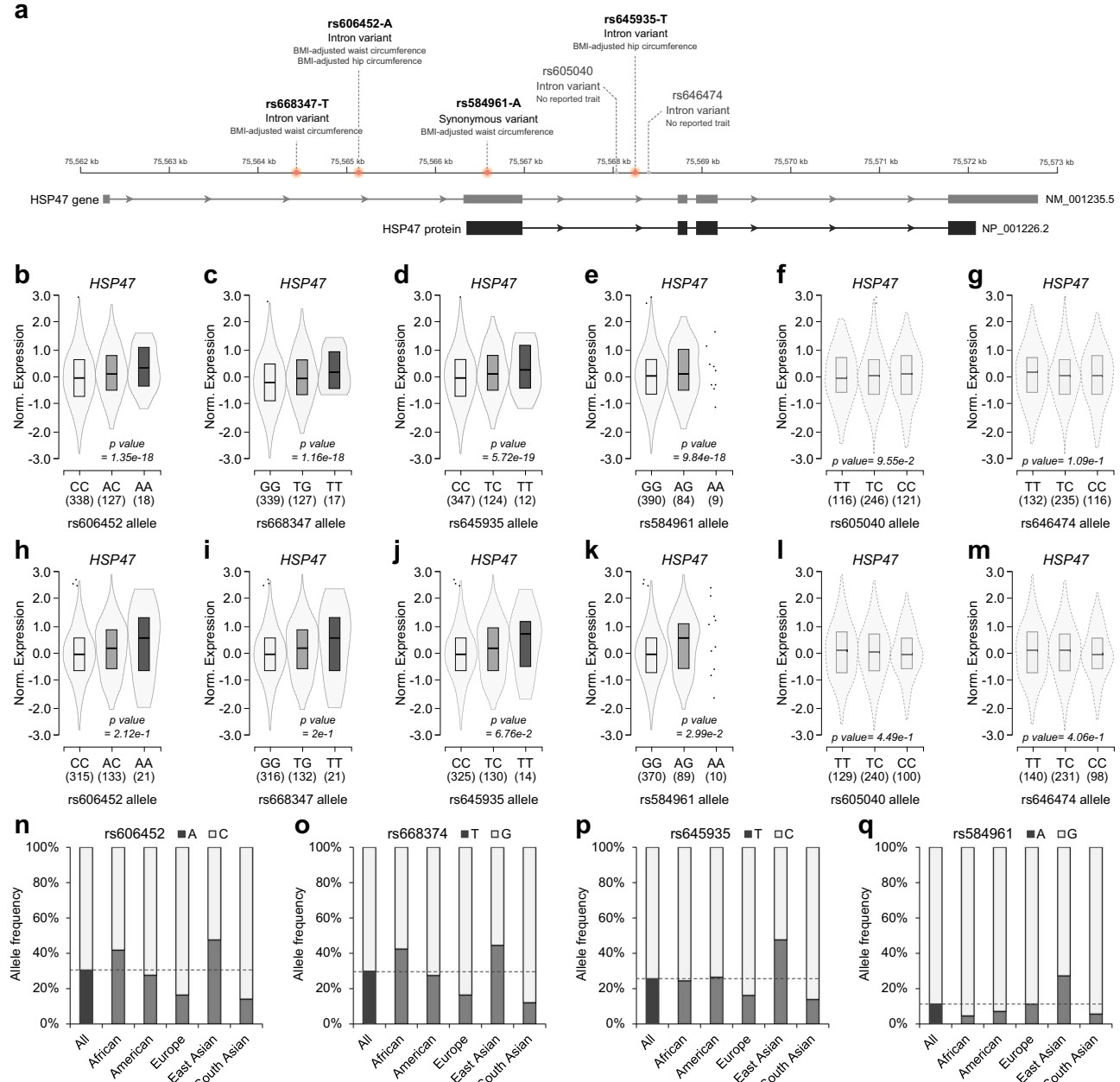

**Fig. 3 | Genetic association of *HSP47* gene expression with body adiposity traits.** **a** Schematic summary of genetic variants in HSP47 gene (NM_001235.5) and protein (NP_001226.2) loci associated with body adiposity traits (rs606452, rs668347, rs645935, and rs584961), BMI-adjusted waist and/or hip circumferences in GWAS catalog, or common variants (rs584961 and rs605040) without any reported traits. eQTL analyses of genetic variants rs606452 (**b**; genotype CC *n* = 338; AC *n* = 127; AA *n* = 18), rs668347 (**c**; genotype GG *n* = 339; TG *n* = 127; TT *n* = 17), rs645935 (**d**; genotype CC *n* = 347; TC *n* = 124; TT *n* = 12), rs584961 (**e**; genotype GG *n* = 390; AG *n* = 84; AA *n* = 9), rs605040 (**f**; genotype TT *n* = 116; TC *n* = 246; CC *n* = 121), and rs646474 (**g**; genotype TT *n* = 132; TC *n* = 235; CC *n* = 116) in cultured human fibroblast cells. eQTL analyses of genetic variants rs606452 (**h**; genotype CC *n* = 315;

AC *n* = 133; AA *n* = 21), rs668347 (**i**; genotype GG *n* = 316; TG *n* = 132; TT *n* = 21), rs645935 (**j**; genotype CC *n* = 325; TC *n* = 130; TT *n* = 14), rs584961 (**k**; genotype GG *n* = 370; AG *n* = 89; AA *n* = 10), rs605040 (**l**; genotype TT *n* = 129; TC *n* = 240; CC *n* = 100), and rs646474 (**m**; genotype TT *n* = 140; TC *n* = 231; CC *n* = 98) in human omental adipose tissue. Allele frequency of rs606452 (**n**), rs668347 (**o**), rs645935 (**p**), and rs584961 (**q**) variants from 1000 genome projects. Statistical *p*-value is from a *t*-test that compares observed normalized enrichment score (NES) from single-tissue eQTL analysis to a null NES of 0. Box plots are shown as median and 25th and 75th percentiles; points are displayed as outliers if they are above or below 1.5 times the interquartile range; *n* refers to sample size. Source data are provided as a Source Data file.

adiposity-related traits, increase the gene expression of *HSP47* gene expression in cultured fibroblast (Fig. 3b–g) and adipose tissues (Fig. 3h–m) with significance or trend toward significance; these *HSP47* gene expression changes were not observed in other tissues, including liver, heart, and pancreas tissue (Supplementary Fig. 4a–l). These results suggest that individuals carrying specific gene variants in the *HSP47* gene loci, including rs606452-A, rs668347-T, rs645935-T, and rs584961-A, exhibit higher gene expression of *HSP47* in fat tissue,

which is associated with increased adiposity, such as waist and/or hip circumference. To gain further insights into the prevalence and widespread nature of these gene variants, we investigated their allele frequencies in the 1000 Genomes population study[29]. *HSP47* gene variants (rs606452-A, rs668347-T, rs645935-T, and rs584961-A) were highly prevalent across various ethnic groups, including African, American, European, East Asian, and South Asian populations (Fig. 3n–q); the average frequencies were rs606452-A (30.4%),

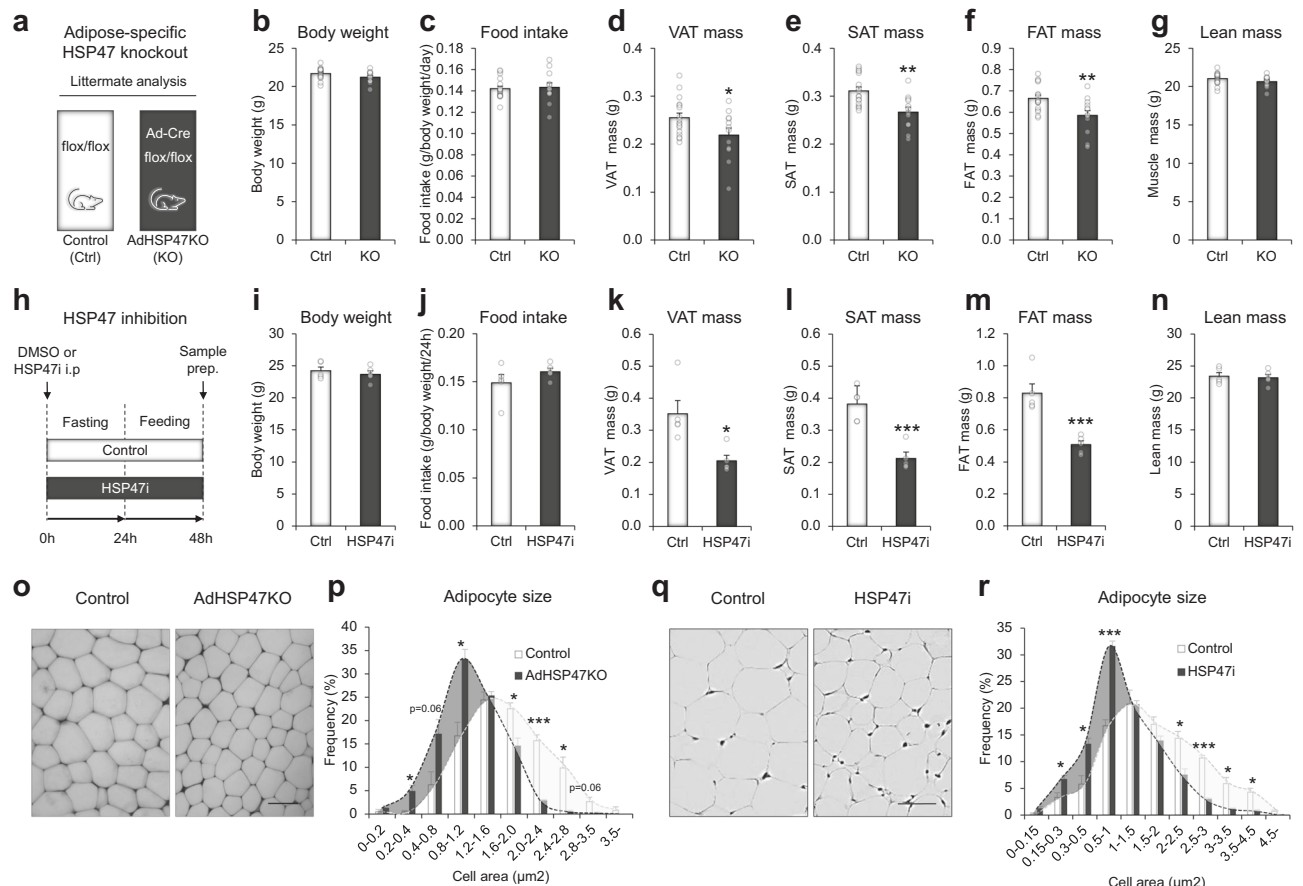

**Fig. 4 | HSP47 ablations reduce the level of body adiposity and adipocyte size.**
**a** Schematic summary of adipose-specific *Hsp47* knockout (AdHSP47KO: KO)
mouse model. Body weight (**b**), food intake (**c**), visceral adipose tissue (VAT; epi-
didymal fat) mass (**d**; *p* = 0.0396), subcutaneous adipose tissue (SAT; inguinal fat)
mass (**e**; *p* = 0.0015), fat (total of VAT, SAT, and BAT) mass (**f**; *p* = 0.004) and lean
mass (**g**) of littermate control and AdHSP47KO mice under normal diet (Control
*n* = 17; AdHSP47KO *n* = 12). **h** Schematic summary of HSP47 inhibition (HSP47i)
mouse model under fasting and refeeding condition. Body weight (**i**), food intake
(**j**), VAT mass (**k**; *p* = 0.0109), SAT mass (**l**; *p* = 0.00039), FAT mass (**m**;
*p* = 0.000842), and lean mass (**n**) of control and HSP47i mice (*n* = 5 each).
**o** Representative whole-mount confocal microscopy image (scale bar indicates
50 μm) of lipid staining in control and AdHSP47KO adipose tissue (epididymal fat).

**p** Quantification of adipocyte size in the adipose tissue (epididymal fat) of control
and AdHSP47KO mice (*n* = 3 each; 0.2–0.4, *p* = 0.027; 0.8–1.2, *p* = 0.0102; 1.6–2.0;
*p* = 0.0184; 2.0–2.4, *p* = 0.00053; 2.4–2.8, *p* = 0.017). **q** Representative hematoxylin
and eosin (H&E) staining image (scale bar 50 μm; grayscale) of control and HSP47i
adipose tissue (epididymal fat). **r** Quantification of adipocyte size in the adipose
tissue (epididymal fat) of control and HSP47i mice (*n* = 3 each; 0.15–0.3, *p* = 0.037;
0.3–0.5, *p* = 0.0168; 0.5–1, *p* = 0.00052; 2–2.5, *p* = 0.01557; 2.5–3, *p* = 0.00012;
3–3.5, *p* = 0.013; 3.5–4.5, *p* = 0.010). Data represent the mean ± SEM. \**p* < 0.05,
\*\**p* < 0.01, and \*\*\**p* < 0.001; *n* refers to sample size. Statistical significance was
determined by two-tailed unpaired *t*-test. Source data are provided as a Source
Data file.

rs668347-T (29.6%), rs645935-T (25.6%), and rs584961-A (11%). These
results suggest that the increased expression of *HSP47* gene by its
specific variants may have a common and worldwide influence on the
increase of body adiposity, contributing to individual diversity.

## HSP47 ablations lower body adiposity in mice
We validated the causal relationship between HSP47 and body adip-
osity in mouse models. First, we generated adipose-specific *Hsp47*
knockout (hereafter AdHSP47KO) mice by crossing *Hsp47*^flox/flox^ mice
with *Adipoq*^Cre^ mice (Fig. 4a). AdHSP47KO mice had no differences in
body weight and food intake (Fig. 4b, c; Supplementary Fig. 5a) but
exhibited lower body adiposity in visceral (epididymal) and sub-
cutaneous (inguinal) fat with no change in lean mass compared with
their littermate control (Fig. 4d–g); there were no significant differ-
ences in the brown fat, muscle and liver mass (Supplementary
Fig. 5b–d). Similar results were observed in female mice; no changes in
body weight and food intake but a significant reduction in visceral
(periovarian) and subcutaneous (inguinal) fat; no differences in brown
fat and muscle mass, but a bit increased liver mass (Supplementary
Fig. 5e–l). Consistent results were obtained under an overeating

condition (7 days of high-fat diet), leading to a significant reduction in
fat tissues (Supplementary Fig. 5m–t). We further assessed the impact
of HSP47 ablation on body adiposity in a non-genetic/pharmacological
model. Col003 (5-benzyl-2-hydroxy-3-nitrobenzaldehyde) is a pre-
viously established cell permeable, small molecule inhibitor of
HSP47[30], which competitively binds to the collagen-binding site on
HSP47, destabilizes the triple helix of collagen, and inhibits its secre-
tion into the extracellular matrix[30,31]. The inhibition of HSP47 by the
specific inhibitor (Fig. 4h; hereafter HSP47i) under fasting/feeding
cycle did not affect body weight and food intake (Fig. 4i, j; Supple-
mentary Fig. 6a), but significantly decreased visceral and sub-
cutaneous fat mass (Fig. 4k–m). HSP47i did not change brown fat and
muscle mass but caused a significant increase in liver mass (Supple-
mentary Fig. 6b–d). Similar results were observed in overfeeding
condition (24 h of fasting followed by 72 h of HFD refeeding) with a
significant reduction in fat tissue mass; there were no changes in
muscle and brown fat mass but an increase in liver mass (Supple-
mentary Fig. 6e–l). The size of adipocyte was significantly reduced in
both AdHSP47KO (Fig. 4o, p) and HSP47i mice (Fig. 4q–r) compared to
the control groups. Collectively, these genetic and non-genetic/

pharmacological ablations of HSP47 in mice clearly demonstrate that HSP47 causally affects body adiposity.

## HSP47 ablations change systemic energy homeostasis

Adipose tissue plays a crucial role in regulating metabolic homeostasis. The increase in liver mass observed in the genetic and/or pharmacological ablation of HSP47 suggested a possible imbalance in energy storage. Therefore, we next investigated the overall effects of HSP47 ablations on systemic energy balance and metabolism using these mouse models. AdHSP47KO mice exhibited higher levels of glycogen and/or triglyceride in the liver compared to the control under both normal and HFD conditions (Supplementary Fig. 7a, b). Similarly, HSP47i mice showed increased glycogen and/or triglyceride accumulation in the liver under normal and/or HFD conditions (Supplementary Fig. 7c, d). Notably, both AdHSP47KO and HSP47i mice exhibited impaired glucose and insulin tolerances with elevated plasma insulin levels (Supplementary Fig. 8a–f). These results are well aligned with previous research that loss of adipose tissue causes glycogen and triglyceride depositions in liver and impairs systemic energy metabolism[32–34]. Collectively, these results suggest that the partial loss of fat tissue by HSP47 ablations led to a shift in energy storage from adipose tissue to liver and impairs systemic energy metabolism.

## HSP47 regulates focal adhesion signaling and PPARγ expression

We next investigated the molecular function of HSP47 in adipocytes associated with body adiposity. HSP47 is a collagen-specific chaperone that controls the folding and secretion of collagen[19,20]. Therefore, we examined the dynamics of collagen protein and its connection with the downstream Integrin/FAK axis and PPARγ, as suggested in the earlier analysis (Fig. 1b, c). Phenotypically, AdHSP47KO mice exhibited a reduced extracellular collagen protein surrounding fat cells (Fig. 5a, b), indicating the role of HSP47 in collagen secretion. The decrease in extracellular collagen matrix was linked with reduced FAK signaling (Phosphorylated FAK; hereafter pFAK) and PPARγ proteins in the adipose tissue of AdHSP47KO mice compared to that of control mice (Fig. 5c–f). Similar results were observed in the adipose tissue of HSP47i mice, showing a significant reduction in pFAK and PPARγ proteins compared to control mice (Fig. 5g–i). Consistent with the results of mouse tissue, the depletion of *Hsp47* by siRNA in 3T3-L1 adipocytes markedly reduced pFAK and PPARγ proteins (Fig. 5j–m), which was associated with a decrease in dimer and trimer forms of collagen protein (Fig. 5n–o), confirming the significant role of HSP47 in collagen protein folding. Similarly, the inhibition of HSP47 in 3T3-L1 adipocytes reduced the interaction of extracellular collagen protein with a major collagen receptor Integrin β1 (Fig. 5p) as determined by co-immunoprecipitation assay, which was associated with decreased pFAK and PPARγ proteins (Fig. 5q–s; Supplementary Fig. 9a–c). Extracellular collagen matrix binds to integrin receptors via short amino acid sequences such as the Arg-Gly-Asp (RGD) motif. The inhibition of extracellular collagen-integrin interactions by synthetic RGD peptides significantly decreased pFAK and PPARγ proteins (Fig. 5t–v). The knockdown of *Itgb1* gene by siRNA in 3T3-L1 showed similar results with a reduction in pFAK and PPARγ proteins (Fig. 5w–z). Collectively, HSP47-linked collagen dynamics, especially the folding, secretion, and interaction with integrin, controls focal adhesion signaling and PPARγ expression in adipocytes.

## HSP47 controls the stability and activity of PPARγ

We next examined the biological relationship between FAK signaling and PPARγ in adipocytes. The direct inhibition of FAK signaling by a specific inhibitor (hereafter FAKi) in 3T3-L1 adipocytes significantly reduced PPARγ protein levels (Fig. 6a, b) without affecting *Pparg* gene expression (Supplementary Fig. 6a). Also, the reduction of PPARγ proteins by genetic and pharmacological ablations of HSP47 in adipocytes and adipose tissue did not change the gene expression of

*Pparg* (Supplementary Fig. 10b–e). These data suggest that HSP47-linked FAK signaling regulates the protein expression of PPARγ. Indeed, when protein synthesis was blocked by cycloheximide (CHX) to evaluate protein stability, FAKi accelerated the decrease of PPARγ protein (Fig. 6c, d). Ubiquitin-linked proteasomal degradation is a well-established pathway for the control of PPARγ protein. The ubiquitin assay of PPARγ protein in HEK293T cells showed the significant ubiquitination of PPARγ protein; the ubiquitin-linked PPARγ reduction by FAKi was restored by co-treatment with MG132, an inhibitor of proteasomal degradation of ubiquitin-conjugated proteins (Fig. 6e). Treatment with MG132 abolished FAKi-mediated decrease of PPARγ protein in 3T3-L1 adipocytes (Fig. 6f). We further searched for putative molecules that mediate FAKi-linked PPARγ degradation. Based on protein interactome profiles of FAK (153 proteins) and PPARγ (282 proteins) in BioGRID, a biomedical interaction repository[35], we found 10 common interacting proteins for both FAK and PPARγ proteins. Among these, MDM2 and STUB1 were identified as well-known E3 ubiquitin ligases, which are key factors for proteasomal degradation (Supplementary Fig. 11a). Co-immunoprecipitation assay in HEK293T cells confirmed the significant binding of PPARγ protein with MDM2, but not with STUB1 (Supplementary Fig. 11b). The knockdown of *MDM2* by siRNA inhibited FAKi-mediated reduction of PPARγ protein in a dose-dependent manner (Supplementary Fig. 11c); the inhibition of MDM2 by the specific inhibitor (hereafter MDM2i) partly restored FAKi-mediated decrease of PPARγ protein not only in HEK293T cells but also in 3T3-L1 adipocytes in a dose-dependent manner (Supplementary Fig. 11d, e; Fig. 6g, h). Similar results were observed with HSP47i in these serial experiments; PPARγ ubiquitin assay in HEK293T cells (Fig. 6i), PPARγ restoration by MG132 (Fig. 6j), and MDM2i (Fig. 6k, i) in 3T3-L1 adipocytes.

Finally, we estimated whether the downregulation of PPARγ protein is responsible for low body adiposity in HSP47 ablations. The target genes of PPARγ, such as *Adipoq*, *Fabp4*, *Cd36*, *Lpl*, *Glut4*, and/or *Scd1*, were significantly decreased in 3T3-L1 adipocytes and adipose tissue of HSP47 ablations without changes in *Pparg* gene expression levels (Fig. 6m, n); *Hsp47* siRNA also significantly reduced PPARγ target genes in 3T3-L1 adipocytes regardless of *Pparg* gene expression (Supplementary Fig. 12a), indicating the reduction of PPARγ activity by HSP47 ablations. To further confirm the contribution of PPARγ activity in the low body adiposity of HSP47 ablations, we utilized a synthetic agonist of PPARγ, pioglitazone. The treatment of pioglitazone in 3T3-L1 adipocytes efficiently restored the reduction of Adiponectin by *Hsp47* siRNA (Supplementary Fig. 12b, c). Of note, the forced activation of PPARγ by pioglitazone (every other day; four times a week administration) fully rescued the mass of white adipose tissue (WAT; epididymal an inguinal fat) in AdHSP47KO mice (Fig. 6o, p). Similarly, the co-treatment of pioglitazone with HSP47i significantly recovered the reduction of WAT (Fig. 6q, r) in alignment with the restoration of Adiponectin in the adipose tissue (Supplementary Fig. 12d, e). Collectively, these data suggest that the lower body adiposity in HSP47 ablations is attributed to the lower stability and activity of PPARγ.

## Discussion

In summary, we found that the expression of HSP47 is abundant in fat tissues, associated with various conditions of high or low body adiposity, and up-and down-regulated by insulin and glucocorticoid hormones, respectively. The increase of HSP47 gene expression by its intron or synonymous variants–frequently found in any ethnic group–are associated with the higher adiposity traits, such as waist and hip circumferences. The ablation of HSP47 by adipose-specific knockout or pharmacological inhibition in mice exhibits lower levels of body adiposity, which were associated with the function of HSP47 regulating the dynamics of collagen protein in fat cells. Mechanistically, HSP47 promotes the folding of collagen protein and its

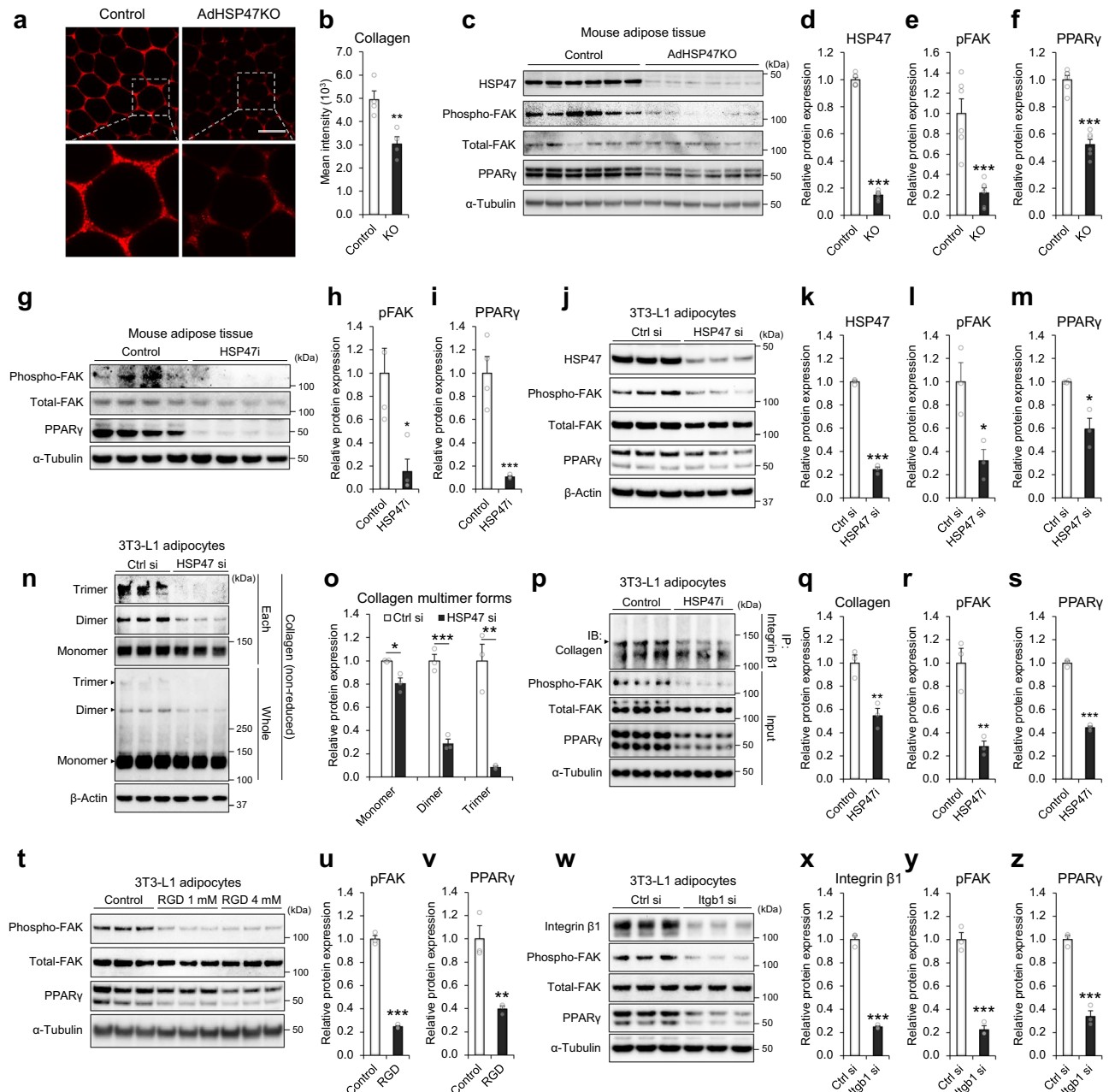

**Fig. 5 | HSP47 ablations decrease FAK signal and PPARγ protein.** Confocal microscopy image (**a**) of extracellular collagen protein (anti-Collagen VI) in the epididymal adipose tissue of control and AdHSP47KO ($n = 4$ each); the arithmetic mean fluorescence intensity for collagen (**b**; $p = 0.0073$). Western blot image (**c**) of HSP47, focal adhesion kinase (FAK) signal, and PPARγ proteins in the epididymal adipose tissue of control and AdHSP47KO ($n = 6$); the densitometry of HSP47 (**d**; $p = 2.96e-12$), phosphorylated FAK (**e**; Phospho-FAK; pFAK; $p = 0.000466$), and PPARγ (**f**; $p = 2.19e-6$) proteins. Western blot image (**g**) of FAK signal and PPARγ proteins in the epididymal adipose tissue of control and HSP47i ($n = 4$ each); the densitometry of pFAK (**h**; $p = 0.011$) and PPARγ (**l**; $p = 0.00074$) proteins. Western blot image (**j**) of HSP47, FAK signal and PPARγ proteins in 3T3-L1 adipocytes after control or *Hsp47* siRNA ($n = 3$ each); the densitometry of HSP47 (**k**; $p = 6.59e-6$), pFAK (**l**; $p = 0.0239$), and PPARγ (**m**; $p = 0.0121$) proteins. Western blot image (**n**) of Collagen multimer forms (monomer, dimer, and trimer; non-reduced; anti-Collagen VI) in 3T3-L1 adipocytes after control or HSP47 siRNA ($n = 3$ each); the

densitometry (**o**) of monomer ($p = 0.0139$), dimer ($p = 0.00043$), and trimer ($p = 0.003$) forms of collagen protein. Western blot image (**p**) of Integrin-bound Collagen (IP; anti-Integrin β1, IB; anti-Collagen VI), FAK signal (Input), and PPARγ (Input) proteins in 3T3-L1 adipocytes after HSP47i for 3 h ($n = 3$ each); the densitometry of Collagen (**q**; anti-Collagen VI; $p = 0.00783$), pFAK (**r**; $p = 0.0058$) and PPARγ (**s**; $p = 1.59e-5$) proteins. Western blot image (**t**) of FAK signal and PPARγ proteins in 3T3-L1 adipocytes after RGD peptide treatment (0, 1, 4 mM) for 16 h ($n = 3$ each); the densitometry of pFAK (**u**; $p = 1.74e-5$) and PPARγ (**v**; $p = 0.0064$) proteins (control vs RGD 4 mM). Western blot image (**w**) of Integrin β1, FAK signal, and PPARγ proteins in 3T3-L1 adipocytes after *Itgb1* siRNA for 48 hours ($n = 3$ each); the densitometry of Integrin β1 (**x**; $p = 2.53e-5$), pFAK (**y**; $p = 0.00035$) and PPARγ (**z**; $p = 0.000361$) proteins. Data represent the mean ± SEM. *$p < 0.05$, **$p < 0.01$, and ***$p < 0.001$; $n$ refers to sample size. Statistical significance was determined by two-tailed unpaired $t$-test. Source data are provided as a Source Data file.

secretion in extracellular matrix enhancing the physical binding with integrin receptor. This interaction between extracellular collagen and subcellular integrin activates FAK signaling, which protects intracellular PPARγ protein from ubiquitin-mediated proteasomal

degradation, partly related to MDM2, and determines the level of body adiposity in alignment with PPARγ activity (Fig. 7; Graphical summary).

Fat tissue is enriched in extracellular components, especially collagens, but the physiological significance, beyond their role as a

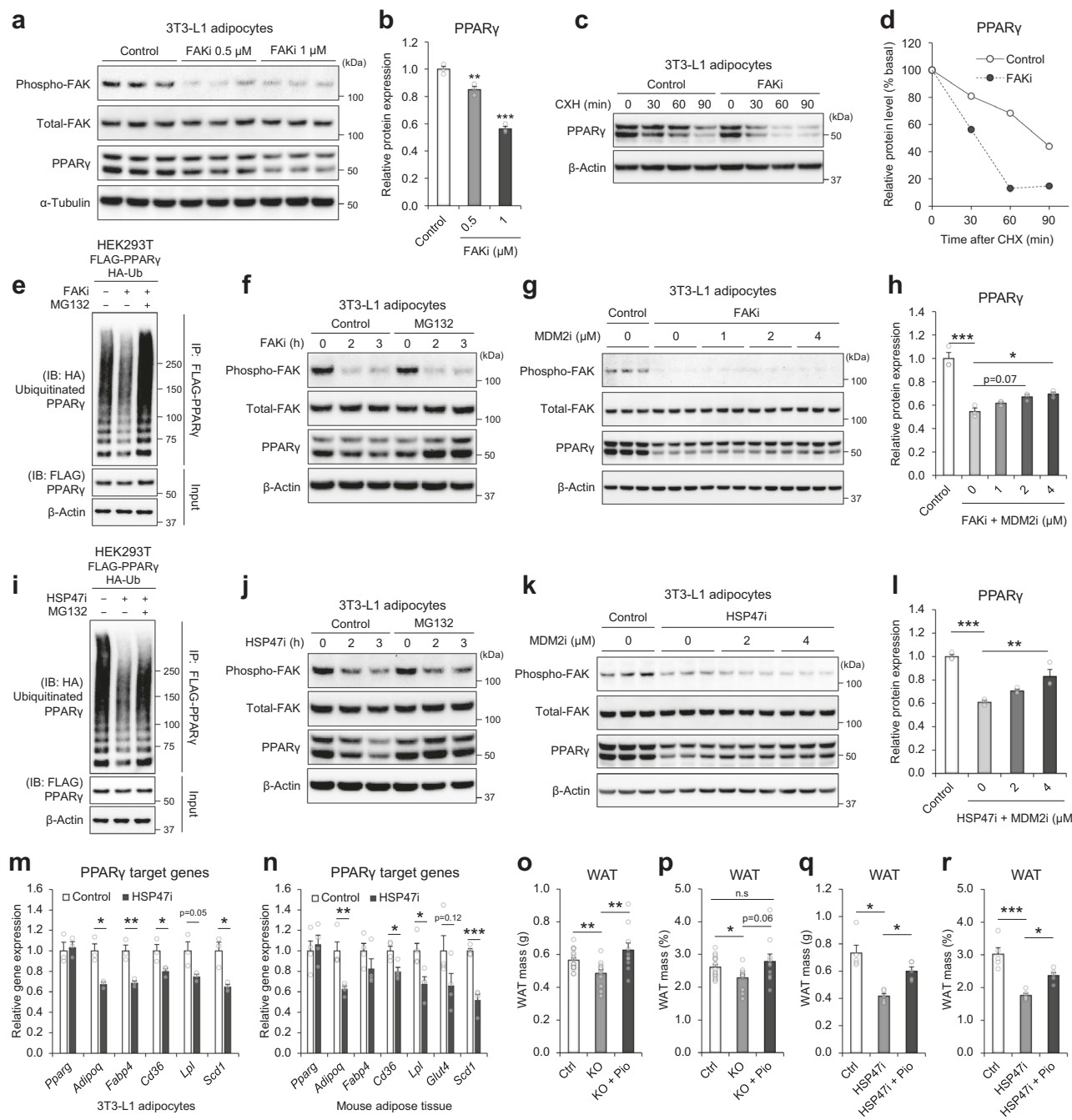

structural or fibrotic factor[36,37], has not been fully explored. In this study, we demonstrate that HSP47 plays a crucial role in controlling the folding of collagen matrix and its secretion in the extracellular matrix to determine the proper size and volume of fat cells. The rearrangement of extracellular collagen matrix changes the dynamics of subcellular focal adhesion and adjusts the intracellular level of PPARγ protein. The connections between extracellular collagen matrix, subcellular focal adhesion, and intracellular PPARγ are directed by HSP47 in fat cells, which plays crucial role in determining the level of body adiposity. These results are consistent with previous studies showing that defects in collagen[38,39], Integrin β1[40], focal adhesion kinase[41], and PPARγ[6–9] cause lower levels of body adiposity and/or lipodystrophy. Collectively, our findings highlight the crucial role of the extracellular collagen matrix, which is linked to the subcellular integrin/FAK axis and intracellular PPARγ protein, not only as an extracellular structure or a mediator of pathological fibrosis, but also

as a regulator of the extra-, sub-, and intracellular components necessary for the proper lipid storage of fat cells.

Body adiposity differs between individuals and depends on a variety of internal and external conditions[14,15]. Although considerable interest and attention have been devoted to understanding the nature of fat tissue, there is still a lack of scientific basis to fully explain individual diversity in body adiposity. In this study, we found that the expression of HSP47 is associated with various conditions of high or low body adiposity–increasing with feeding, overeating, and obesity, whereas decreasing with fasting, exercise, calorie restriction, bariatric surgery, and cachexia. Based on its expression profiles associated with systemic nutrition/energy status, we defined that the expression of HSP47 is tightly regulated by insulin and glucocorticoid hormones. Moreover, we found the genetic relevance of HSP47 showing that higher HSP47 gene expression, as driven by its intronic (rs606452, rs668347, and rs645935) or synonymous (rs584961) variants, was

**Fig. 6 | The low body adiposity of HSP47 ablations is attributed to the decreased stability and activity of PPARγ.** Western blot image (**a**) of FAK signal and PPARγ proteins in 3T3-L1 adipocytes treated with FAK inhibitor (PF-573228; FAKi; 0.5 and 1 μM; $n = 3$ each) for 3 h; the densitometry graph (**b**; control vs 0.5 μM, $p = 0.0063$; control vs 1 μM, $p < 0.0001$) of PPARγ protein. Western blot image (**c**) of focal adhesion kinase (FAK) signal in 3T3-L1 adipocytes after FAKi (1 μM) treatment with cycloheximide (50 ug/ml) for 0, 30, 60, and 90 min; the densitometry graph (**d**) of PPARγ protein. **e** Western blot image of ubiquitinated PPARγ protein in FLAG-PPARγ and HA-Ub overexpressed HEK293T cells after FAKi (10 μM) treatment with/ without MG132 (10 μM) for 3 h. **f** Western blot image of FAK signal and PPARγ proteins in 3T3-L1 adipocytes after FAKi (1 μM) treatment with/without MG132 (10 μM) for 0, 2 and 3 h. Western blot image (**g**) of FAK signal and PPARγ proteins in 3T3-L1 adipocytes after FAKi (1 μM) treatment with/without MDM2i (MI-773; 0, 1, 2, and 4 μM) for 3 hours; the densitometry graph (**h**; Control vs FAKi, $p < 0.0001$; FAKi vs FAKi + MDM2i 4 μM, $p = 0.0289$) of PPARγ protein. **i** Western blot image of ubiquitinated PPARγ protein in FLAG-PPARγ and HA-Ub overexpressed HEK293T cells after HSP47i (200 μM) treatment with/without MG132 (10 μM) for 3 h. **j** Western blot image of FAK signal and PPARγ proteins in 3T3-L1 adipocytes after HSP47i (200 μM) treatment with/without MG132 (10 μM) for 0, 2 and 3 h. Western blot image (**k**) of FAK signal and PPARγ proteins in 3T3-L1 adipocytes after HSP47i (200 μM) treatment with/without MDM2i (MI-773; 0, 2, and 4 μM) for 3 h; the densitometry graph

(**l**; Control vs HSP47i, $p = 0.0002$; HSP47i vs HSP47i + MDM2i 4 μM, $p = 0.0073$) of PPARγ protein. **m** Realtime qPCR data of PPARγ and the target genes (*Adipoq*, $p = 0.01045$; *Fabp4*, $p = 0.007259$; *Cd36*, $p = 0.048212$; *Lpl*, $p = 0.051$; *Scd1*, $p = 0.016136$) in 3T3-L1 adipocytes after HSP47i (200 μM) treatment for 3 h ($n = 3$ each). **n** Realtime qPCR data of PPARγ and the target genes (*Adipoq*, $p = 0.0065$, *Cd36*, $p = 0.017$; *Lpl*, $p = 0.0196$; *Scd1*, $p = 0.000272$) in in vivo mouse adipose tissue (epididymal fat) after HSP47i (100 mg/kg; single shot followed by 24 h fasting and 24 h refeeding) treatment ($n = 4$ each). White adipose tissue (WAT) mass (**o** the total mass of epididymal and inguinal fat; **p** WAT percentage per body weight) of control (Ctrl) and AdHSP47KO (KO) with/without pioglitazone (pio; 15 mg/kg; every other day for a week) treatment (Ctrl $n = 17$; KO $n = 12$; KO + Pio $n = 10$). The total mass of WAT, Ctrl vs KO, $p = 0.0096$; KO vs KO + Pio, $p = 0.0068$. WAT percentage, Ctrl vs KO, $p = 0.0395$. **q, r** White adipose tissue (WAT) mass (**q**; total mass of WAT, **r**; WAT percentage) of control (Ctrl) and HSP47i (100 mg/kg) with/without pioglitazone (pio; 30 mg/kg) treatment ($n = 5$ each). WAT total mass, Ctrl vs HSP47i, $p = 0.0122$; HSP47i vs HSP47i + Pio, $p = 0.0122$. WAT percentage, Ctrl vs HSP47i, $p < 0.0001$; HSP47i vs HSP47i + Pio, $p = 0.0189$. Data represent the mean ± SEM. *$p < 0.05$, **$p < 0.01$, and ***$p < 0.001$; $n$ refers to sample size. Statistical significance was determined by Tukey–Kramer test (**h** and **l**), two-tailed unpaired *t*-test (**m** and **n**), and Wilcoxon test (**o–r**). Source data are provided as a Source Data file.

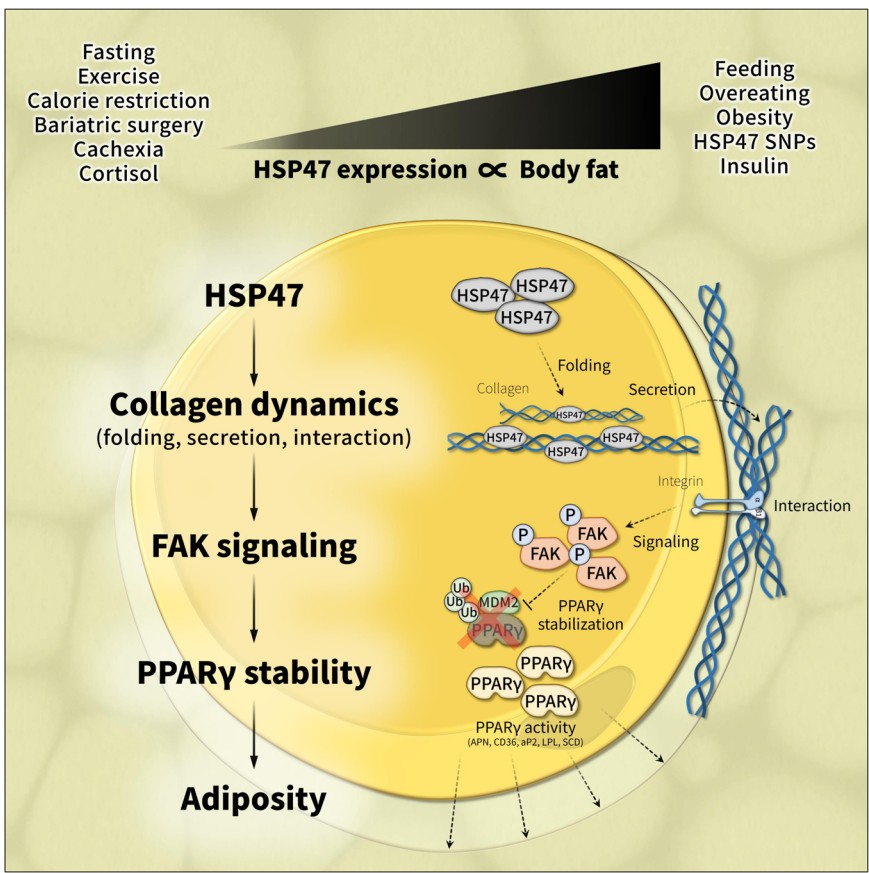

**Fig. 7 | Graphical summary.** The level of body fat in individuals is subject to considerable variation and influenced by numerous factors, such as physiological, pathological, environmental, hormonal, and genetic conditions. In this study, we have identified HSP47, a collagen-specific chaperone, as a key determinant of body adiposity that is abundant in fat tissues and strongly correlated with human conditions of high or low body adiposity. HSP47 expression is increased by feeding, overeating, and obesity, while it is decreased by fasting, exercise, calorie restriction, bariatric surgery, and cachexia. It is also significantly correlated with several body adiposity traits, including body fat mass, BMI, waist, and hip circumferences. Additionally, insulin and glucocorticoids, endocrine hormones that are associated with systemic nutritional states and body fat storage/loss, respectively up- and

down-regulate the expression of HSP47. In humans, increased HSP47 gene expression resulting from intron or synonymous variants is associated with higher body adiposity traits, such as hip and waist circumferences. Similarly, in mice, the ablation of HSP47 results in significantly lower body adiposity compared to the control group. Mechanistically, HSP47 plays a critical role in promoting collagen protein dynamics, including folding, secretion, and interaction with integrin. This, in turn, activates FAK signaling and preserves PPARγ protein from proteasomal degradation, partly related to MDM2. Our findings emphasize the critical role of HSP47 in determining body adiposity, providing valuable insights into the individual variability and differences in body adiposity traits observed in diverse circumstances.

associated with higher body adiposity traits. We believe that the molecular features of HSP47, which carry physiological, pathological, environmental, hormonal, and genetic associations with fat cells, could help explain individual traits related to body adiposity, or serve as a useful indicator of an individual's state of body adiposity.

Lipid storage is a fundamental and evolutionarily conserved feature found in diverse organisms from yeast to humans, which offers a metabolic advantage for adaptability to variable nutrient environments. Adipose tissue is a specially evolved organ in vertebrates that efficiently stores energy as triglyceride within lipid droplets. It dynamically adjusts its size and mass in response to systemic energy supply and demand, regulating the systemic energy balance and metabolism. In our study, we found that genetic and pharmacological ablations of HSP47 in mice led to a partial loss of fat tissue, accompanied by ectopic energy deposition in the liver and impaired glucose and insulin tolerances. These findings are consistent with numerous basic and clinical reports that loss of fat tissue redirects energy deposition towards non-adipose organs, particularly increasing the hepatic glycogen and triglyceride contents, and causes glucose intolerance and insulin resistance, and ultimately the development of diabetes[32–34].The study could serve as a scientific foundation that offers a molecular mechanism of proper fat tissue expansion and its association with systemic energy balance. Further clinical investigations are necessary to fully understand the relationship between HSP47 defects and metabolic disturbances.

This study is initiated and based on human data, which is a strength of this study, demonstrating significant associations of HSP47 and body adiposity traits. The findings are supported by transcriptome data from both human and rodent species, as well as in vivo and in vitro experiments, which collectively suggest that HSP47 plays a role in determining body adiposity. Furthermore, a previous proteomic study showed that the protein expression of HSP47 were regulated in relation to adipocytes size[42], which is well aligning with this current research. We acknowledge that HSP47 is not the sole factor influencing body adiposity, as other obesogenic elements, such as other genetic variants[18], inflammation[43], oxidative stress[44], adipocytokines[45], and energy metabolism[2], comprehensively contribute to the overall level of body adiposity. To validate the attributes of human HSP47, we conducted experiments using simplified genetic and pharmacological mouse models under normal or short-term high-fat diet conditions. Further research incorporating modified conditions, interventions, and longer time periods may be necessary to gain additional insights. Careful titration would be necessary in any clinical and/or pharmacological applications of HSP47 inhibition, considering the potential implications for metabolic homeostasis and overall health.

In conclusion, this study sheds new light on the previously unexplored nature of HSP47 in determining body adiposity. We identified its significant associations with body adiposity, hormonal regulations, genetic involvements, metabolic impacts, key factors, and molecular mechanisms that explain how our body determines body adiposity. We hope that this study will contribute to the broader scientific community and advance our understanding of the complex interplay between genetic, environmental, and lifestyle factors in regulating body adiposity.

## Methods
### Materials
All experimental materials used in this study were listed in Supplementary Table 1.

### Animals
All mouse studies were approved by the Ethics Review Committee for Animal Experimentation of Osaka University, Graduate School of Medicine, and conducted in accordance with the Osaka University Institutional Animal Care and Use Committee Guidelines. The mice were housed in groups of one to three or five mice per cage, maintained in a room with a controlled temperature ($23 \pm 1.5\,°C$) and humidity ($45 \pm 15\%$) on a 12-h light/dark cycle, and provided with free access to water and chow (MF; Oriental Yeast, Tokyo, Japan). All mouse experiments were performed under the background of C57BL/6J mice. C57BL/6J mice were obtained from Charles River Japan (Yokohama, Japan) and acclimated for at least one week before the experiment. Male C57/BL6J mice (8–9 weeks old) were used to determine the tissue distribution of the HSP47 protein in the epididymal fat (visceral adipose tissue; VAT), inguinal fat (subcutaneous adipose tissue; SAT), interscapular brown fat (brown adipose tissue; BAT), liver, gastrocnemius muscle, pancreas, kidney, brain, heart, and lung. For the explant experiments, epididymal fat from 8–9-week-old C57BL/6J mice was excised, cut into even 3–5 mm sizes, cultured in DMEM (high glucose) with 10% FBS and penicillin/streptomycin, and used for the corresponding purpose. To perform in vivo acute inhibition of the insulin receptor, C57/BL6J mice (8-9 weeks old) were treated with the insulin receptor inhibitor OSI-906 (IRi; 100 mg/kg) and had free access to water but not chow. Twenty-four hours after treatment, the epididymal fat was excised and analyzed for the corresponding experiment purpose. Adiponectin-Cre mice were provided by E. Rosen (Beth Israel Deaconess Medical Center). *Nr3c1* (*Gr*; glucocorticoid receptor) floxed mice were purchased from the Jackson Laboratory (stock no. 021021) and were previously published in our lab[46]. To generate adipocyte-specific GR knockout (AGRKO) mice, *Gr* floxed mice were crossed with Adiponectin-Cre mice. Corticosterone was added to the drinking water of 8–13 weeks old of Gr$^{flox/flox}$ control or AGRKO mice (final concentration, 50 µg/mL) for two weeks, and the epididymal fat were excised and analyzed for the corresponding experiment purpose. *Serpinh1* (*Hsp47*) floxed mice were previously established by Kazuhiro Nagata lab[47] and purchased from RIKEN BioResource Research Center (RIKEN BRC; BRC No. RBRC10972). To generate adipocyte-specific HSP47 knockout (AdHSP47KO), *Hsp47* floxed mice were crossed with Adiponectin-Cre mice. Male *Hsp47*$^{flox/flox}$ control and AdHSP47KO mice (7–8 weeks old; littermate) were used for the analysis. Pioglitazone (Pio; 15 mg/kg) was administered every other day for a week by intraperitoneal (i.p.) injection, adjusted to the final endpoint of analysis. For the pharmacological inhibition of HSP47, HSP47i (Col003; 100 mg/kg) was administered to C57/BL6J mice (9 weeks old) by i.p. injection, with/without Pioglitazone (Pio; 30 mg/kg), followed by a 24-h fasting and 24-h refeeding intervention. High-fat diet (60 kcal%; Research Diets, D12492) was utilized for overeating models. The epididymal fat, inguinal fat, interscapular brown fat, liver, and gastrocnemius muscle tissues were rapidly harvested after anesthesia and promptly frozen in liquid nitrogen. For the intraperitoneal insulin tolerance test (ITT), mice (4–7 months old) were fasted for 5 h and then injected with 0.5 units/kg body weight (BW) of insulin. In the intraperitoneal glucose tolerance test (GTT), mice (4-7 months old) were also fasted for 5 h before the injection of glucose at a dose of 1 g/kg BW. Glucose levels were measured at the indicated times using a portable glucose meter (Glutest Neo alpha; Sanwa Kagaku Kenkyusho, Nagoya, Japan) through tail vein sampling. Adiponectin-Cre, *Hsp47*, and *Gr* floxed mice were genotyped according to the supplier's genotyping protocols.

### Cells
HEK293T cells and 3T3-L1 mouse fibroblasts were obtained from ATCC and cultured in DMEM (high glucose) supplemented with 10% FBS and penicillin/streptomycin. To differentiate 3T3-L1 cells into adipocytes, differentiation medium containing 3-isobutyl-1-methylxanthine (0.5 mmol/L), dexamethasone (1 µmol/L), and insulin (1 µmol/L) were used. The cells were used in experiments 7 days after differentiation. All HSP47i and RNAi experiments were conducted after completing the differentiation of adipocytes, specifically between Day 7 and Day 9 following the induction of adipocyte differentiation. The cells were analyzed up to 3 h after

HSP47i treatment or 48 h after HSP47 siRNA treatment in adipocytes. For the cycloheximide chase assay, 3T3-L1 adipocytes were treated with cycloheximide (50 ug/ml) at corresponding time points, with or without other experimental procedures, and were subjected to western blotting.

## Small Interfering RNA (siRNA)

The differentiated 3T3-L1 adipocytes (day 5–7) or HEK293T cells in 10-cm dish were treated with 2.5 ml of trypsin-EDTA and incubated at 37 °C for 2 min. The cells were washed in a 50 mL conical centrifuge tube and centrifuged at 500 g for 5 min. In the meantime, the siRNA mixture of Opti-MEM, siRNA solution, and RNAiMAX was prepared according to the manufacturer's instruction. The cell pellet was gently resuspended in the culture medium ($10^6$ cells/mL), plated onto 12-well dish with the siRNA mixture, and incubated for 2 days. The protein and/or RNA of the cells were extracted and analyzed in accordance with the experiment purpose.

## Western blot

Cultured cells or tissue samples were lysed in RIPA buffer with protease and phosphatase inhibitor cocktails. Protein concentration was determined by the bicinchoninic acid method. The protein solution was diluted in SDS sample buffer solution with or without DTT supplementation and heat processing (5 min at 95 °C). SDS samples were separated by SDS-PAGE, transferred to polyvinylidene difluoride (0.2 μm PVDF) membrane, blocked by Tris-buffered saline (137 mmol/L NaCl, 20 mmol/L Tris–HCl, pH 7.6) containing 0.05% Tween-20 (TBS-T) and 5% skim milk for one hour at room temperature. The membranes were then incubated with primary antibodies in TBS-T and 5% skim milk overnight at 4 °C. After washing with TBS-T (10 min, three times), the membranes were incubated with enhanced chemiluminescence horseradish peroxidase–linked secondary antibodies in TBS-T and 5% skim milk for one hour at room temperature. After washing with TBS-T (10 min, three times), the immunoreactive bands were visualized by Pierce Western Blotting Substrate Plus. Chemiluminescent images were captured using the ChemiDoc Touch imaging system (BIO-RAD). Quantification was conducted by densitometry using ImageJ software.

## Immunoprecipitation

Cells were lysed with TNE buffer (10 mM Tris-HCl, 150 mM NaCl, 1 mM EDTA, 1% NP40) containing proteinase and phosphatase inhibitors. For the immunoprecipitation of FLAG-tagged protein, anti-FLAG M2 Affinity Gel was used to immunoprecipitate the cells. The immunoprecipitated cells were washed with TNE buffer and then eluted with 200 μg/mL FLAG peptide. For the immunoprecipitation of Integrin β1, the extracellular proteins of 3T3-L1 adipocytes were cross-linked by DTSSP (3,3´-dithiobis[sulfosuccinimidylpropionate]) according to the manufacturer's instructions. The cross-linked proteins were then lysed with TNE buffer and immunoprecipitated using Dynabeads Protein G (Sigma-Aldrich). The immunoprecipitated proteins were eluted by SDS sample buffer and subjected to western blotting.

## Cell-based ubiquitination assay

HEK293T cells were transfected with pRK5-HA-Ubiquitin-WT (Addgene, 17608; HA-Ub) and pcDNA3.1-FLAG-PPARγ plasmids using Lipofectamine 3000 reagent according to the manufacturer's instructions. 24 h after transfection, the cells were treated with MG132 (10 μM) with FAKi or HSP47i for 3 h. The cells were lysed by boiling in a buffer containing 2% SDS, 150 mM NaCl, 10 mM Tris-HCl, and 1 mM DTT. These lysates were diluted 5-fold in dilution buffer containing 150 mM NaCl, 10 mM Tris-HCl, and 1% Triton X-100, and then immunoprecipitated with anti-FLAG M2 Affinity Gel. The immunoprecipitates were washed four times with dilution buffer, eluted with 200 μg/mL FLAG peptide, and subjected to western blotting.

## RNA Isolation and RT-qPCR

Total RNA was isolated from cells or tissues using TRI reagent according to the manufacturer's instructions. The prepared RNA samples were reverse transcribed using Transcriptor Universal cDNA Master (Roche). Real-time quantitative PCR (RT-qPCR) was performed using FastStart Essential DNA Green Master (Roche) and the LightCycler® 96 System (Roche), following the manufacturer's instructions. Rplp0 (36B4) or Cyclophilin was used as an internal control. The primers used in this study are listed in Supplementary Table 1.

## Whole-mount immunofluorescence

Adipose tissues were cut into 2.5 mm × 2.5 mm pieces and fixed with 4% paraformaldehyde in PBS overnight at 4 °C. After briefly washing with TBS (Tris-buffered saline) for 1 min, the tissue samples were incubated with 0.1% TBST (TBS with 0.1% Tween) containing 10% goat serum (blocking buffer) overnight at 4 °C. Subsequently, the tissue samples were incubated with appropriate primary antibodies in blocking buffer for 1–1.5 h at room temperature or overnight at 4 °C. After washing with 0.1% TBST (four times for 30 min each), the tissue samples were incubated with secondary antibodies in blocking buffer for 2 h. After washing with 0.1% TBST (four times for 30 min each), the tissue samples were transferred to a glass bottom dish, treated with 35 μl of Fluoromount-G buffer, and dried overnight at room temperature in an upside-down position. Fluorescence was then observed under a confocal microscope (LSM880; ZEISS).

## Immunohistochemistry

The adipose tissues were excised, fixed in 10% formalin, and subsequently embedded in paraffin. Sections were then prepared and processed for haematoxylin and eosin staining.

## Liver triglyceride and glycogen assay

Triglycerides and glycogen levels in the liver were quantified using the Triglyceride E-test (Wako Pure Chemical Industries) and Glycogen Assay Kit (Cayman), respectively, following the manufacturers' recommended protocols.

## Plasma insulin measurement

Plasma insulin was measured by using Mouse Insulin ELISA Kit (MS303; Morinaga, Yokohama, Japan) according to the instructions provided by the manufacturer.

## Transcriptome datasets analysis

All transcriptome datasets used in this study are listed in Supplementary Table 1. Analyses were performed using the GEO2R analysis tool or Microsoft Excel. For the adipose enrichment analysis, median gene expression data of human tissues was obtained from GTEx. The gene expression of adipose tissues (the average of subcutaneous and omentum adipose tissue) was divided by the total gene expression of 54 different tissue samples to calculate the adipose enrichment score.

## Gene Ontology and Pathway analysis

Gene Ontology (GO) and Kyoto Encyclopedia of Genes and Genomes (KEGG) pathway analyses were performed using the ToppGene web-based analysis software.

## Gene variant eQTL and population analysis

The eQTL (expression Quantitative Trait Locus) analysis of corresponding gene variants was performed using the web-based eQTL analysis software (eQTL Dashboard). The detailed information on the analysis process is described on the GTEx Portal methods page (https://www.gtexportal.org/home/methods). The gene code for

HSP47 and variant IDs for rs606452, rs668347, rs645935, rs584961, rs605040, and rs646474 were obtained through a search on the GTEx portal. The allele frequency of HSP47 variants in 1000 genomes projects were downloaded from Ensemble population genetics (https://asia.ensembl.org/).

## Statistical analysis

All biological results were repeated at least twice in similar and/or same experiments to confirm reproducibility. Significance within and between groups was analyzed using paired or unpaired Student's $t$ tests in Excel software. Significance among multiple groups was analyzed using Tukey–Kramer tests or Wilcoxon tests in JMP Pro 15.2.1 software. A $p$-value of less than 0.05 was considered statistically significant. All data are presented as mean ± s.e.m.

## Reporting summary

Further information on research design is available in the Nature Portfolio Reporting Summary linked to this article.

# Data availability

All information and accession codes for transcriptome datasets used in this study are listed in Supplementary Table 1 and available on NCBI GEO website under accession codes GSE154612; GSE28005; GDS3602; GSE92405; GSE116801; GSE77962; GSE72158; GSE20571; GSE70353; GDS3142; GDS4918; GDS5824; GDS4811; GDS2319; GSE60596; GSE70857; GDS4899; GDS3135; GDS2946; GSE58575; GSE118978; GSE176295; GDS2956; GSE35581; GSE88966. All data and materials are available in the paper and the supplementary information. Source data are provided with this paper.

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

## Acknowledgements
This work was partly supported by the Japan Society for the Promotion of Science Grant-in-aid for Scientific Research (grant no. 22K16413) and Front Runner of Future Diabetes Research (FFDR). We thank all members of the department of Metabolic Medicine at Graduate School of Medicine, Osaka University.

## Author contributions
All authors made direct and/or intellectual contributions to the work and approved it for publication. J.S. designed the study and researched the data. J.S. and S.T. performed mouse experiments. J.S. and I.S. wrote the manuscript. Y.O. and R.H generated and managed AGRKO mice. S.I. and K.N. provided AdHSP47KO mice and discussed the research. H.S. performed cell preparation and mouse genotyping. S.T., Y.O, R.H., H.S., S.N., T.O., S.K. (Sachiko Kobayashi), H.N., S.K. (Shunbun Kita), O.M., and A.F. contributed to the discussion and reviewed the manuscript. J.S. is the guarantor of this work and, as such, had full access to all the data in the study and takes responsibility for the integrity of the data and the accuracy of the data analysis.

## Competing interests
The authors declare no competing interests.
