## [Peer Review File · Nature Communications]

HSP47 determines the amount of body fatREVIEWER COMMENTS

Reviewer #1 (Remarks to the Author):

Shin et al. took an integrative approach by combining bioinformatic approaches such as public dataset such as GWAS catalog, GEO and GTEx and biological analyses such as genetically engineered animal models, cell culture experiments and chemical inhibitors and they elucidated the role of collagen-specific chaperone, HSP47 in the regulation of body fatness. The experiments were carefully designed and conducted well, the message is clear, and the conclusion was well supported by the experimental data with a mechanistic insight that involves regulation of the stability of PPAR γ by HSP47.

I have a few comments.

Major comments:

1 What are the overall effects of either genetic or pharmacological inhibition on glucose and lipid metabolism, in the context of a normal chow condition or high-fat diet-induced obesity? This is important information considering the relatively acute pharmacological action of the HSP47 inhibitor, that is, nearly 50% reduction of white adipose tissue masses in 48 hours (Figure 4). The action could be either beneficial or detrimental to metabolic homeostasis. Supposing that both Ctrl and KO animals eat an equal amount of food, where do the ingested lipid go? Does it accumulate in the liver? Or was it burned within either white or brown adipose tissues? If the fat accumulates in the liver as suggested by the authors (extended figure 4), simple suppression of HSP47 may not be beneficial and careful titration may be necessary. Does accumulation of fat in the liver also occur in the adipose KO mice (either heterozygous or homozygous)?

2 Please provide more detailed information as to the mode of action of the HSP47 inhibitor Col003, its chemical structure, its basic effect on collagen synthesis, the specificity of the inhibition, potential off-target and adverse effect, etc...

3 Figure 3. GWAS and eQTL data suggested the examined polymorphisms were associated with the expression of HSP47.

3.1 Was the eQTL regulation specifically seen in adipose tissue and fibroblasts or was it also seen in other types of tissues?

3.2 What was the direction of the regulation of HSP gene expression and adiposity by the HSP polymorphisms? Was the risk alleles for increased body weight associated with increased HSP gene expression?

Minor comments:

4 3T3-L1 experiments. It is not clear exactly how inhibition of HSP47 by HSP47i and RNAi was conducted in terms of the timing of the treatment. It is not clear if HSP47i alters the extent of differentiation of 3T3-L1.

Reviewer #2 (Remarks to the Author):

In the present study, HSP47, a collagen-specific chaperone, is identified as a key factor in body fatness. The expression of HSP47 in adipose tissue was abundant; it increased with feeding, overeating, and obesity; it decreased with fasting, exercise, calorie restriction, and bariatric surgery; and it correlated with fat mass, BMI, waist, and hip circumferences. HSP47 expression was up- and down-regulated by insulin and glucocorticoids, respectively.

Increased expression of HSP47 intron or synonymous variants in humans was associated with greater body fatness. Adipose-specific knockouts or pharmacological inhibitions of HSP47 led to lower body fatness in mice than in controls. Collagen dynamics are enhanced by HSP47 through fold, secretion, and interaction with integrins. This induce FAK signaling and protects PPAR γ protein from proteasomal degradation, partly related to MDM2. HSP47 plays a significant role in determining body fatness individually and in various circumstances. However, there some concerns that need to be addressed:

Introduction:

1-Adipose tissue, composed of adipocytes, is a specialized organ that stores energy as a fat, commonly referred to as fat tissue or fat cell¹. I suggest removing “fat cell” because the adipose tissue is referred to as fat tissue itself. There are at least two classes of fat cells—white and brown.

2-In general, the introduction section should be rewritten. Detailed information related to the state of the art is missing. The authors focused on obesity. However, the consequences

of obesity at the molecular level, especially in adipose tissue should be considered. Adipose tissue, once considered merely an energy storage depot, is now recognized as a metabolically active and highly influential organ. In obesity, adipose tissue undergoes extensive remodeling, leading to altered adipokine secretion, inflammation, and perturbations in adipocyte function. It is within this dynamic tissue that the molecular consequences of obesity are most pronounced. This tissue plays a key role in the development of systemic metabolic dysfunction. Besides, the gene HSP47, should be introduced correctly. Indeed, it is known that Hsp47 is regulated in adipose and this was described using a proteomics approach in isolated adipocytes from adipose-specific insulin receptor knock-out (PMID: 15131120).

Results:

Fig. 1: We performed gene ontology (GO) analysis on the top 500 genes and found that components of collagen-containing extracellular matrix (ECM) and focal adhesion were enriched in fat tissue (Fig. 1b), suggesting that these structures may play an important role in adipose tissue function. Indeed, ECM and focal adhesion play an important role in adipose tissue function. I would remove "suggesting". These results are confirming previous findings. Fig.2a and Fig.2.b: Why did the authors use different insulin and Dexa concentrations to test Hsp47 gene expression?

Fig. 4: Adipose-specific HSP47 knockout exhibits lower fat mass, both VAT and SAT. Based on these results, I would recommend either indirect calorimetry (energy expenditure and RER) and/or exploring other metabolic tissues (triglycerides in liver) to elucidate where the energy in the KO mice is going compared to WT mice. Besides, since insulin mediates HSP47, metabolic phenotyping (GTT, ITT...) is highly recommended as well. Did the authors check something in female mice? HFD feeding?

REVIEWER COMMENTS

Reviewer #1 (Remarks to the Author):

Shin et al. took an integrative approach by combining bioinformatic approaches such as public dataset such as GWAS catalog, GEO and GTEx and biological analyses such as genetically engineered animal models, cell culture experiments and chemical inhibitors and they elucidated the role of collagen-specific chaperone, HSP47 in the regulation of body fatness. The experiments were carefully designed and conducted well, the message is clear, and the conclusion was well supported by the experimental data with a mechanistic insight that involves regulation of the stability of PPAR γ by HSP47.

I have a few comments.

Major comments:

1. What are the overall effects of either genetic or pharmacological inhibition on glucose and lipid metabolism, in the context of a normal chow condition or high-fat diet-induced obesity? This is important information considering the relatively acute pharmacological action of the HSP47 inhibitor, that is, nearly 50% reduction of white adipose tissue masses in 48 hours (Figure 4). The action could be either beneficial or detrimental to metabolic homeostasis. Supposing that both Ctrl and KO animals eat an equal amount of food, where do the ingested lipid go? Does it accumulate in the liver? Or was it burned within either white or brown adipose tissues? If the fat accumulates in the liver as suggested by the authors (extended figure 4), simple suppression of HSP47 may not be beneficial and careful titration may be necessary. Does accumulation of fat in the liver also occur in the adipose KO mice (either heterozygous or homozygous)?

Answer

We appreciate the reviewer for the insightful comments. In response to the reviewer's suggestions, we conducted a comprehensive investigation to explore the overall effects of genetic and pharmacological ablations of HSP47 on glucose and lipid metabolism in relevant mouse models. As predicted by the reviewer, the reduction in fat tissue mass was accompanied by the ectopic accumulation of glucose and lipid in the liver in the form of glycogen and/or triglycerides; adipose-specific knockout of HSP47 (AdHSP47KO) exhibited higher glycogen and/or triglyceride contents in the liver under both normal chow and/or high-fat diet conditions (Figure to reviewer 1-1. a and c). Similar results were observed in the pharmacological model as well; the inhibition of HSP47 (HSP47i) led to an increase in glycogen and/or triglyceride contents in the liver under normal and/or high-fat diet conditions (Figure to reviewer 1-1. c and d).

Figure to reviewer 1-1. Increased glycogen and triglyceride contents in liver of AdHSP47KO and HSP47i mice. a-b, Measurement of glycogen (a) and triglyceride (b) contents in liver of AdHSP47KO mice under normal (n=8 each) or 7 days of high-fat diet (n=6 each) condition. c-d, Measurement of glycogen (a) and triglyceride (b) contents in liver of HSP47i mice under normal or high-fat diet condition (24 hours of fasting followed by 24 hours of normal diet, n= 5 each, or 72 hours of high-fat diet refeeding, n=6 each, with or without HSP47 inhibitor). Data represent the mean \pm SEM. #p<0.25, *p < 0.05, **p < 0.01, and ***p < 0.001.

Furthermore, the reduced adipose capacity to store energy in adipose and ectopic glucose and fat deposition in liver by AdHSP47KO and HSP47 were associated with impaired glucose and insulin tolerances with elevated plasma insulin levels (Figure to reviewer 1-2).

Figure to reviewer 1-2. Impairment of glucose and insulin tolerance in AdHSP47KO and HSP47i mice. a and b, Glucose (a) and insulin (b) tolerance test in control and AdHSP47KO mice (control, Ctrl, n=4; AdHSP47KO, KO, n=5). c, Plasma insulin level in control and AdHSP47KO fasted after 5 hours fasting (n=8 each). d and e, Glucose (d) and insulin (e) tolerance test in control and HSP47i mice (n=8 each). f, Plasma insulin level in control and HSP47i mice after 5 hours fasting (n=5 each). Data represent the mean \pm SEM. #p<0.25, *p < 0.05, **p < 0.01, and ***p < 0.001.

These results are well aligned with previous research that partial and/or complete loss of adipose tissue, termed lipodystrophy, causes glycogen and triglyceride depositions in liver and impairs systemic energy metabolism²¹⁻²³. Taken together, these results indicate that both pharmacological and genetic ablation of HSP47 induce a redirection of energy deposition towards non-adipose organs, particularly the liver, and impair systemic energy metabolism. As the reviewer mentioned, careful titration would be necessary in any clinical and/or pharmacological applications of HSP47 inhibition, considering the potential implications for metabolic homeostasis and overall health.

We included these new findings in the revised figure and manuscript as follows:

------(Page 8; Line 253-267)

HSP47 ablations change systemic energy homeostasis.

Adipose tissue plays a crucial role in regulating metabolic homeostasis. The increase in liver mass observed in the genetic and/or pharmacological ablation of HSP47 suggested a possible imbalance in energy storage. Therefore, we next investigated the overall effects of HSP47 ablations on systemic energy balance and metabolism using these mouse models. AdHSP47KO mice exhibited higher levels of glycogen and/or triglyceride in the liver compared to the control under both normal and HFD conditions (Extended Fig. 7a and 7b). Similarly, HSP47i mice showed increased glycogen and/or triglyceride accumulation in the liver under normal and/or HFD conditions (Extended Fig. 7c and 7d). Notably, both AdHSP47KO and HSP47i mice exhibited impaired glucose and insulin tolerances with elevated plasma insulin levels (Extended Fig. 8a-8f). These results are well aligned with previous research that loss of adipose tissue causes glycogen and triglyceride depositions in liver and impairs systemic energy metabolism²¹⁻²³. Collectively, these results suggest that the partial loss of fat tissue by HSP47 ablations led to a shift in energy storage from adipose tissue to liver and impairs systemic energy metabolism.

------(Page 12; Line 387-399; discussion)

Lipid storage is a fundamental and evolutionarily conserved feature found in diverse organisms from yeast to humans, which offers a metabolic advantage for adaptability to variable nutrient environments. Adipose tissue is a specially evolved organ in vertebrates that efficiently stores energy as triglyceride within lipid droplets. It dynamically adjusts its size and mass in response to systemic energy supply and demand, regulating the systemic energy balance and metabolism. In our study, we found that genetic and pharmacological ablations of HSP47 in mice led to a partial loss of fat tissue, accompanied by ectopic energy deposition in the liver and impaired glucose and insulin tolerances. These findings are consistent with numerous basic and clinical reports that loss of fat tissue leads to imbalanced energy accumulation in other organs, especially liver²¹⁻²³. The study could serve as a scientific foundation that offers a molecular mechanism of proper fat tissue expansion and its association with systemic energy balance. Further clinical investigations are necessary to fully understand the relationship between HSP47 defects and metabolic disturbances.

2. Please provide more detailed information as to the mode of action of the HSP47 inhibitor Col003, its chemical structure, its basic effect on collagen synthesis, the specificity of the inhibition, potential off-target and adverse effect, etc...

Answer

Thank you for your valuable feedback. In response to the reviewer's suggestion, we included more detailed information about Col003. Col003 is a small molecule inhibitor that competitively binds to the collagen-binding site on HSP47. This binding interaction causes a reduction in the stability of the collagen triple helix and consequently inhibits the secretion of collagen into the extracellular matrix (ECM). The inhibitory effect of Col003 specifically targets collagen synthesis and does not impact other ECM proteins, such as laminin.

Originally identified as a potential lead compound for fibrosis treatment, Col003 has demonstrated promising inhibitory effects on collagen synthesis. However, given the findings of the current research, it is imperative to exercise caution in order to prevent interference with metabolic homeostasis. Further experimental and clinical investigations are required to ascertain the suitable dosage and treatment duration for potential clinical applications.

We incorporated the information in the revised manuscript with proper references as follows:

------(Page 8; Line 236-241)

Col003 (5-benzyl-2-hydroxy-3-nitrobenzaldehyde) is a previously established cell permeable, small molecule inhibitor of HSP47³⁵, which competitively binds to the collagen-binding site on HSP47, destabilizes the triple helix of collagen, and inhibits its secretion into the extracellular matrix^{35,36}.

----- (Page 12; Line 413-415)

Careful titration would be necessary in any clinical and/or pharmacological applications of HSP47 inhibition, considering the potential implications for metabolic homeostasis and overall health.

Thank you once again for your valuable input, which has greatly enhanced the comprehensiveness of our manuscript.

3 Figure 3. GWAS and eQTL data suggested the examined polymorphisms were associated with the expression of HSP47.

3.1 Was the eQTL regulation specifically seen in adipose tissue and fibroblasts or was it also seen in other types of tissues?

Answer

Thank you for the important comments. HSP47 gene variants, including rs606452-A, rs668347-T, rs645935-T, and rs584961-A, increase its gene expression in adipose tissue and fibroblast cells. However, there were no significant differences in other tissues, such as liver, heart, and pancreas (Figure to reviewer 1-3).

Figure to reviewer 1-3. eQTL analysis of HSP47 genetic variants in liver, heart and pancreas. a-d, eQTL analyses of genetic variants rs606452 (a), rs668347 (b), rs645935 (c), rs584961 (d) in human liver tissue. e-h, eQTL analyses of genetic variants rs606452 (e), rs668347 (f), rs645935 (g), and rs584961 (h) in human heart tissue. i-l, eQTL analyses of genetic variants rs606452 (i), rs668347 (j), rs645935 (k), and rs584961 (l) in human pancreas tissue. Box plots are shown as median and 25th and 75th percentiles; points are displayed as outliers if they are above or below 1.5 times the interquartile range.

We incorporated the information in the revised manuscript as follows:

----- (Page 7; Line 209-211)

; these *HSP47* gene expression changes were not observed in other tissues, including liver, heart, and pancreas tissue (Extended Fig. 4a-4l).

3.2 What was the direction of the regulation of HSP gene expression and adiposity by the HSP polymorphisms? Was the risk alleles for increased body weight associated with increased HSP gene expression?

Answer

We thank the reviewer for comments and apologize for the insufficient explanation regarding the connection between the risk alleles and the direction of body fatness traits. The genetic variants (rs606452-A, rs668347-T, rs645935-T, and rs584961-A) that lead to increased gene expression of HSP47 are linked with higher waist and hip circumferences. In revising the manuscript, we've taken care to avoid any potential confusion regarding the phenotypic directions of body fatness traits and HSP47 expression as follow:

------(Page 7; Line 196)

Paragraph title

Genetic increases of HSP47 gene associate with higher body fatness

------(Page 7; Line 211-222)

These results suggest that individuals carrying specific gene variants in the HSP47 gene loci, including rs606452-A, rs668347-T, rs645935-T, and rs584961-A, exhibit higher gene expression of HSP47 in fat tissue, which is associated with increased adiposity, such as waist and/or hip circumference. To gain further insights into the prevalence and widespread nature of these gene variants, we investigated their allele frequencies in the 1000 Genomes population study³⁴. HSP47 gene variants (rs606452-A, rs668347-T, rs645935-T, and rs584961-A) were highly prevalent across various ethnic groups, including African, American, European, East Asian, and South Asian populations (Fig. 3n-3q); the average frequencies were rs606452-A (30.4%), rs668347-T (29.6%), rs645935-T (25.6%), and rs584961-A (11%). These results suggest that the increased expression of HSP47 gene by its specific variants may have a common and worldwide influence on the increase of body fatness, contributing to individual diversity.

Minor comments:

4 3T3-L1 experiments. It is not clear exactly how inhibition of HSP47 by HSP47i and RNAi was conducted in terms of the timing of the treatment. It is not clear if HSP47i alters the extent of differentiation of 3T3-L1.

Answer

Thank you for the comments. We apologize for not providing sufficient explanation regarding the timing of HSP47 inhibition by HSP47i and RNAi in the 3T3-L1 experiments. All HSP47i and RNAi experiments were conducted after completing the differentiation of adipocytes, specifically between Day 7 and Day 9 following the induction of adipocyte differentiation. We analyzed the samples up to 3 hours after HSP47i treatment or 48 hours after HSP47 siRNA treatment in adipocytes and the effects of HSP47 inhibition occur relatively acute (within 3 hours for HSP47i) or at an early stage (48 hours for HSP47 siRNA) in adipocytes, ensuring that the experiments are not associated with the process of adipogenesis. In this revised version, we have provided a clear and detailed description of these experimental procedures to avoid any misunderstandings or concerns in the revised version as below.

------(Page 14; Line 483-486)

All HSP47i and RNAi experiments were conducted after completing the differentiation of adipocytes, specifically between Day 7 and Day 9 following the induction of adipocyte differentiation. The cells were analyzed up to 3 hours after HSP47i treatment or 48 hours after HSP47 siRNA treatment in adipocytes.

We appreciate the reviewer again for the opportunity to clarify these important details.

Reviewer #2 (Remarks to the Author):

In the present study, HSP47, a collagen-specific chaperone, is identified as a key factor in body fatness. The expression of HSP47 in adipose tissue was abundant; it increased with feeding, overeating, and obesity; it decreased with fasting, exercise, calorie restriction, and bariatric surgery; and it correlated with fat mass, BMI, waist, and hip circumferences. HSP47 expression was up- and down-regulated by insulin and glucocorticoids, respectively. Increased expression of HSP47 intron or synonymous variants in humans was associated with greater body fatness. Adipose-specific knockouts or pharmacological inhibitions of HSP47 led to lower body fatness in mice than in controls. Collagen dynamics are enhanced by HSP47 through fold, secretion, and interaction with integrins. This induce FAK signaling and protects PPAR γ protein from proteasomal degradation, partly related to MDM2. HSP47 plays a significant role in determining body fatness individually and in various circumstances. However, there some concerns that need to be addressed:

Introduction:

1-Adipose tissue, composed of adipocytes, is a specialized organ that stores energy as a fat, commonly referred to as fat tissue or fat cell¹. I suggest removing "fat cell" because the adipose tissue is referred to as fat tissue itself. There are at least two classes of fat cells—white and brown.

Answer

We appreciate the reviewer's comments. As per the suggestion, we removed the term "fat cell" as below. Thank you for bringing this to our attention.

------(Page 3; Line 56)

Adipose tissue, composed of adipocytes, is a specialized organ to store energy as fat, thus commonly referred to as fat tissue¹.

2-In general, the introduction section should be rewritten. Detailed information related to the state of the art is missing. The authors focused on obesity. However, the consequences of obesity at the molecular level, especially in adipose tissue should be considered. Adipose tissue, once considered merely an energy storage depot, is now recognized as a metabolically active and highly influential organ. In obesity, adipose tissue undergoes extensive remodeling, leading to altered adipokine secretion, inflammation, and perturbations in adipocyte function. It is within this dynamic tissue that the molecular consequences of obesity are most pronounced. This tissue plays a key role in the development of systemic metabolic dysfunction. Besides, the gene HSP47, should be introduced correctly. Indeed, it is known that Hsp47 is regulated in adipose and this was described using a proteomics approach in isolated adipocytes from adipose-specific insulin receptor knock-out (PMID: 15131120).

Answer

Thank you for the comments. I agree with the reviewer's comments that the introduction part is missing detailed information related to the state of the art on adipose tissue. As suggested by the reviewer, we rewritten the introduction with more detailed and recent information on adipose tissue and cited the relevant paper. The revised manuscripts are as follow.

------(Page 3-4; Line 55-103)

Adipose tissue, composed of adipocytes, is a specialized organ to store energy as fat, thus commonly referred to as fat tissue¹. It is distributed throughout the body, including subcutaneous fat beneath the skin, visceral fat around internal organs, and brown fat in the back, neck, and shoulder area². Fat tissue can dynamically change its size and mass in response to nutritional and hormonal states. During eating or overeating conditions, increased blood insulin levels stimulate glucose uptake and lipid accumulation in fat cells, promoting the expansion of fat tissue³. Conversely, fasting or starving conditions induce glucocorticoid hormone, which triggers the lipolysis of stored fat, leading to a reduction in fat tissue⁴. The peroxisome proliferator-activated receptor- γ (PPAR γ) is a master regulator of fat cells⁵. Defects in PPAR γ result in the loss of fat tissue⁶⁻⁹, while its activation by agonists, such as pioglitazone, promotes lipid storage and the expansion of fat tissue^{10,11}. The unique properties of fat tissue and its regulation by various nutritional, hormonal, and molecular factors have been well established, yet the intricate connections between these elements is not fully understood.

Body fatness, which refers to the amount or percentage of body fat, varies greatly between individuals, ranging from as low as 5% in very thin/lean people to over 40% in the cases of morbid obesity¹². Additionally, each person exhibits a unique susceptibility to gain body fat under regular eating or overeating conditions¹³. These individual variations are influenced by a combination of physiological, environmental, pathological, and genetic factors. Physiologically, eating habits and patterns, which affect food calorie intake, fasting duration, and hormone levels, have a significant impact on body fatness^{14,15}. Environmentally,

engaging in high physical activity or exercise can also contribute to lower body fatness^{14,15}. Pathologically, obesity is characterized by high levels of body fat tissue¹⁵, while cachexia, a wasting disorder, leads to a significant loss of fat tissue¹⁶. Bariatric surgery has been shown effective results in reducing body fatness in obese patient¹⁷. Genetic factors, such as single nucleotide polymorphisms (SNPs), also play a role in determining body fatness traits, such as fat mass, BMI, waist circumference, and hip circumferences¹⁸. Despite the comprehensive contribution of physiological, pathological, environmental, and genetic backgrounds to body fatness, a unified molecular basis for its regulation remains elusive.

Adipose tissue, once considered merely an energy storage depot, is now recognized as a metabolically active and highly influential organ. It plays a crucial role in regulating systemic energy balance and metabolic homeostasis; thus, the maladaptation of fat tissue can lead to various metabolic disturbances, such as ectopic energy storage, glucose intolerance, impaired insulin sensitivity, and diabetes mellitus¹⁹⁻²³. In the case of obesity, the excessive accumulation of fat causes adipocyte dysfunction with altered adipokine secretion and inflammation, contributing to the development of metabolic disorders, such as hepatic steatosis, insulin resistance, and type 2 diabetes^{19,20}. Notably, these metabolic abnormalities are not exclusive to obesity; they can also occur in cases of fat tissue deficiency or absence, known as lipodystrophy²¹⁻²³. The partial or complete loss of fat tissue redirects energy deposition towards non-adipose organs, particularly increasing the hepatic glycogen and triglyceride contents²¹⁻²³. Moreover, systemic energy imbalance leads to glucose intolerance and insulin resistance, and ultimately the development of diabetes²¹⁻²³. Despite these significant effects, the molecular mechanism underlying proper fat tissue expansion and the regulation of systemic energy balance remain largely unknown.

HSP47 is a collagen-specific molecular chaperone, which plays a critical role in collagen folding and secretion^{24,25}. In this study, we identify HSP47 as a significant determinant of body fatness. Through a series of *in silico*, *in vivo* and *in vitro* analyses, we elucidate how HSP47 determines body fatness, along with its regulation, metabolic impact, and molecular basis.

------(Page 12; Line 404-405)

Furthermore, a previous proteomic study showed that the protein expression of HSP47 were regulated in relation to adipocytes size⁴², which is well aligning with this current research.

Results:

Fig. 1: We performed gene ontology (GO) analysis on the top 500 genes and found that components of collagen-containing extracellular matrix (ECM) and focal adhesion were enriched in fat tissue (Fig. 1b), suggesting that these structures may play an important role in adipose tissue function. Indeed, ECM and focal adhesion play an important role in adipose tissue function. I would remove “suggesting”. These results are confirming previous findings.

Answer

We appreciate the reviewer's comments. As suggested by the reviewer, we removed the “suggesting” part (Page 5; Line 114).

Fig.2a and Fig.2.b: Why did the authors use different insulin and Dexa concentrations to test Hsp47 gene expression?

Answer

Thank you for the thorough feedback. In response to the reviewer's comments, we carried out a series of experiments to assess the impact of insulin and dexamethasone dosages on HSP47 expression in 3T3-L1 adipocytes and *ex vivo* adipose explants. Both insulin and dexamethasone showed dose-dependent regulation of HSP47 within the 0-100 nM range in 3T3-L1 adipocytes (Figure to reviewer 2-1. a and b). Similarly, we found dose-dependent regulatory effects on HSP47 expression by both insulin and dexamethasone within the range of 0 to 100 nM concentrations in *ex vivo* adipose explants, but the effectiveness of the dose-dependent HSP47 regulation was either minimal or slightly attenuated at a dosage of 1000 nM. We have incorporated these titration results into the extended figure (Figure to reviewer 2-1. c and d; Extended Figure3). Additionally, we have replaced Fig. 2b with the outcomes from the newly conducted adipose explant experiment, utilizing a concentration of 100 nM (Figure to reviewer 2-1 e). This alignment with Fig. 2a aims to enhance the clarity and consistency of our presentation. Once again, thank you for your insightful comments, which have undoubtedly contributed to the comprehensiveness of our study.

Figure to reviewer 2-1. a and b, Relative protein expression and western blot image of HSP47 in 3T3-L1 adipocytes after 48 hours of insulin (a; ins; 0, 1, 10, and 100 nM) or dexamethasone (b; Dex 0, 1, 10, and 100 nM) treatment. c and d, Relative protein expression and western blot image of HSP47 in *ex vivo* adipose explant after 48 hours of insulin (c; ins; 0, 1, 10, 100, and 1000 nM) or dexamethasone (d; Dex 0, 1, 10, 100, and 1000 nM) treatment. e, Relative protein expression and western blot image of HSP47 in *ex vivo* cultured mouse adipose tissue after insulin (100 nM) or dexamethasone (100 nM) treatments for 48 hours (n=4 each). Data represent the mean \pm SEM. ***p < 0.001.

Fig. 4: Adipose-specific HSP47 knockout exhibits lower fat mass, both VAT and SAT. Based on these results, I would recommend either indirect calorimetry (energy expenditure and RER) and/or exploring other metabolic tissues (triglycerides in liver) to elucidate where the energy in the KO mice is going compared to WT mice. Besides, since insulin mediates HSP47, metabolic phenotyping (GTT, ITT...) is highly recommended as well. Did the authors check something in female mice? HFD feeding?

Answer

We appreciate the reviewer for the insightful comments. In response to the reviewer's suggestions, we conducted a comprehensive investigation to explore the overall effects of genetic and pharmacological ablations of HSP47 on glucose and lipid metabolism in relevant mouse models. As predicted by the reviewer, the reduction in fat tissue mass was accompanied by the ectopic accumulation of glucose and lipid in the liver in the form of glycogen and/or triglycerides; adipose-specific knockout of HSP47 (AdHSP47KO) exhibited higher glycogen and/or triglyceride contents in the liver under both normal chow and/or high-fat diet conditions (Figure to reviewer 2-2. a and c). Similar results were observed in the pharmacological model as well; the inhibition of HSP47 (HSP47i) led to an increase in glycogen and/or triglyceride contents in the liver under normal and/or high-fat diet conditions (Figure to reviewer 2-2. c and d).

Figure to reviewer 2-2. Increased glycogen and triglyceride contents in liver of AdHSP47KO and HSP47i mice. a-b, Measurement of glycogen (a) and triglyceride (b) contents in liver of AdHSP47KO mice under normal (n=8 each) or 7 days of high-fat diet (n=6 each) condition. c-d, Measurement of glycogen (a) and triglyceride (b) contents in liver of HSP47i mice under normal or high-fat diet condition (24 hours of fasting followed by 24 hours of normal diet, n= 5 each, or 72 hours of high-fat diet refeeding, n=6 each, with or without HSP47 inhibitor). Data represent the mean ± SEM. #p < 0.25, *p < 0.05, **p < 0.01, and ***p < 0.001.

Furthermore, the reduced adipose capacity to store energy in adipose and ectopic glucose and fat deposition in liver by HSP47 ablations, both AdHSP47KO and HSP47i, were associated with impaired glucose and insulin tolerances with elevated plasma insulin levels (Figure to reviewer 2-3).

Figure to reviewer 2-3. Impairment of glucose and insulin tolerance in AdHSP47KO and HSP47i mice. a and b, Glucose (a) and insulin (b) tolerance test in control and AdHSP47KO mice (control, Ctrl, n=4; AdHSP47KO, KO, n=5). c, Plasma insulin level in control and AdHSP47KO fasted after 5 hours fasting (n=8 each). d and e, Glucose (d) and insulin (e) tolerance test in control and HSP47i mice (n=8 each). f, Plasma insulin level in control and HSP47i mice after 5 hours fasting (n=5 each). Data represent the mean \pm SEM. #p<0.25, *p<0.05, **p<0.01, and ***p<0.001.

These results are well aligned with previous research that partial and/or complete loss of adipose tissue, termed lipodystrophy, causes glycogen and triglyceride depositions in liver and impairs systemic energy metabolism²¹⁻²³. Taken together, these results indicate that both pharmacological and genetic ablation of HSP47 induce a redirection of energy deposition towards non-adipose organs, particularly the liver, and impair systemic energy metabolism.

As suggested by the reviewer, we investigated on HSP47 in female. First, we estimated the expression level of *HSP47* in the adipose tissue of female subjects compared to male subjects. There were no expressional differences between males and females in subcutaneous and visceral adipose tissues (Figure to reviewer 2-3. a). Furthermore, we found similar instances of obesity-induced upregulation of the *HSP47* gene in both male and female adipocytes.

Figure to reviewer 2-3. HSP47 gene expression in human male and female adipose tissue and adipocytes. a, Gene expression (TPM) of *HSP47* in human subcutaneous (SAT; male n=445, female n=218) and visceral adipose tissue (VAT; male n=371, female 170) from male and female subjects (GTEX). b, Relative gene expression of *HSP47* in human lean and obese subcutaneous adipocytes from male and female subjects (GDS3602; n=5 each). Data represent the mean \pm SEM. *p < 0.05.

We also investigated the impact of AdHSP47KO in female mice. Consistent with the results in male mice, female AdHSP47KO mice showed no differences in body weight and food intake. However, they exhibited lower body fatness in visceral (periovarian fat) and subcutaneous (inguinal fat) adipose tissues compared to their littermate controls. There were no changes in brown fat and muscle mass, but a slight increase in liver mass was observed in female mice without change in lean mass (Figure to reviewer 2-4).

Figure to reviewer 2-4. lower body fatness in female AdHSP47KO. a-h, Body weight (a), food intake (b), visceral adipose tissue (c; VAT, periovarian fat), subcutaneous adipose tissue (h; SAT; inguinal fat), brown adipose tissue (d; BAT), skeletal muscle (e), liver (f), lean (g) mass of female control (n=8) and AdHSP47KO mice (n=7). Data represent the mean \pm SEM. #p < 0.25, *p < 0.05 and ***p < 0.001.

As suggested by the reviewer, we estimated these genetic and pharmacological model mice under high-fat diet conditions. Female AdHSP47KO exhibited no changes in body weight and food intake but a significant reduction in visceral (periovarian) and subcutaneous (inguinal) fat. There were no differences in brown fat and muscle mass, but a bit increased liver mass (Figure to reviewer 2-5. a-h). Consistent results were obtained under 7 days of high-fat diet condition, leading to a significant reduction in fat tissues (Figure to reviewer 2-5. i-p).

Figure to reviewer 2-5. lower body fitness in AdHSP47KO and HSP47 mice under short-term high fat diet condition. a-h, Body weight (a), food intake (b), visceral adipose tissue (c; VAT, epididymal fat), subcutaneous adipose tissue (d; SAT; inguinal fat), brown adipose tissue (e; BAT), skeletal muscle (f), liver (g), lean (h) mass of male control and AdHSP47KO mice under 7 days of high-fat diet (n=7 each). i-p, Body weight (i), food intake (j), visceral adipose tissue (k; VAT, periovarian fat), subcutaneous adipose tissue (l; SAT; inguinal fat), brown adipose tissue (m; BAT), skeletal muscle (n), liver (o), lean (p) mass of control and HSP47i mice (n=5 each) under 3 days of high-fat diet refeeding after 24 hours fasting. Data represent the mean \pm SEM. #p<0.25, *p<0.05 and **p<0.01.

We've incorporated these new results in the revised figure and manuscript as follows:

------(Page 6; Line 181-182)

these effects exhibited dose dependency within the range of 0 to 100 nM (Extended Fig. 3a-3d).

------(Page 8; Line 253-267)

HSP47 ablations change systemic energy homeostasis.

Adipose tissue plays a crucial role in regulating metabolic homeostasis. The increase in liver mass observed in the genetic and/or pharmacological ablation of HSP47 suggested a possible imbalance in energy storage. Therefore, we next investigated the overall effects of HSP47 ablations on systemic energy balance and metabolism using these mouse models. AdHSP47KO mice exhibited higher levels of glycogen and/or triglyceride in the liver compared to the control under both normal and HFD conditions (Extended Fig. 7a and 7b). Similarly, HSP47i mice showed increased glycogen and/or triglyceride accumulation in the liver under normal and/or HFD conditions (Extended Fig. 7c and 7d). Notably, both AdHSP47KO and HSP47i mice exhibited impaired glucose and insulin tolerances with elevated plasma insulin levels (Extended Fig. 8a-8f). These results are well aligned with previous research that loss of adipose tissue causes glycogen and triglyceride depositions in liver and impairs systemic energy metabolism²¹⁻²³. Collectively, these results suggest that the partial loss of fat tissue by HSP47 ablations led to a shift in energy storage from adipose tissue to liver and impairs systemic energy metabolism.

------(Page 8; Line 231-248)

Similar results were observed in female mice; no changes in body weight and food intake but a significant reduction in visceral (periovarian) and subcutaneous (inguinal) fat; no differences in brown fat and muscle mass, but a bit increased liver mass (Extended Fig. 5e-5l). Consistent results were obtained under an overeating condition (7 days of high-fat diet), leading to a significant reduction in fat tissues (Extended Fig.5m-5t). We further assessed the impact of HSP47 ablation on body fatness in a non-genetic/pharmacological model. Col003 (5-benzyl-2-hydroxy-3-nitrobenzaldehyde) is a previously established cell permeable, small molecule inhibitor of HSP47³⁵, which competitively binds to the collagen-binding site on HSP47, destabilizes the triple helix of collagen, and inhibits its secretion into the extracellular matrix^{35,36}. The inhibition of HSP47 by the specific inhibitor (Fig. 4h; hereafter HSP47i) under fasting/feeding cycle did not affect body weight and food intake

(Fig. 4i and 4j; Extended Fig. 6a), but significantly decreased visceral and subcutaneous fat mass (Fig. 4k-4m). HSP47i did not change brown fat and muscle mass but caused a significant increase in liver mass (Extended Fig. 6b-6d). Similar results were observed in overfeeding condition (24 hours of fasting followed by 72 hours of HFD refeeding) with a significant reduction in fat tissue mass; there were no changes in muscle and brown fat mass but an increase in liver mass (Extended Fig. 6e-6l).

------(Page 12; Line 387-399; discussion)

Lipid storage is a fundamental and evolutionarily conserved feature found in diverse organisms from yeast to humans, which offers a metabolic advantage for adaptability to variable nutrient environments. Adipose tissue is a specially evolved organ in vertebrates that efficiently stores energy as triglyceride within lipid droplets. It dynamically adjusts its size and mass in response to systemic energy supply and demand, regulating the systemic energy balance and metabolism. In our study, we found that genetic and pharmacological ablations of HSP47 in mice led to a partial loss of fat tissue, accompanied by ectopic energy deposition in the liver and impaired glucose and insulin tolerances. These findings are consistent with numerous basic and clinical reports that loss of fat tissue leads to imbalanced energy accumulation in other organs, especially liver²¹⁻²³. The study could serve as a scientific foundation that offers a molecular mechanism of proper fat tissue expansion and its association with systemic energy balance. Further clinical investigations are necessary to fully understand the relationship between HSP47 defects and metabolic disturbances.

REVIEWERS' COMMENTS

Reviewer #1 (Remarks to the Author):

I think the concerns I raised were appropriately addressed by the authors in the revised manuscript.

Reviewer #2 (Remarks to the Author):

the authors have addressed the comments properly and now the manuscript has been significantly improved. However, there are 2 minor points to be addressed:

TITLE: the current title is a bit confused. HSP47 does not determine the body fatness itself.

The authors demonstrated that adiposity regulates gene expression of HSP47. I would suggest something like this: "HSP47 levels determine the degree of body adiposity".

Introduction: line 84-98. The information about adipose tissue is redundant.

Reviewer #1 (Remarks to the Author):

I think the concerns I raised were appropriately addressed by the authors in the revised manuscript.

Author's response:

We genuinely appreciate your time and thoughtful suggestions, which have greatly enriched the manuscript.

Reviewer #2 (Remarks to the Author):

the authors have addressed the comments properly and now the manuscript has been significantly improved. However, there are 2 minor points to be addressed:
TITLE: the current title is a bit confused. HSP47 does not determine the body fatness itself. The authors demonstrated that adiposity regulates gene expression of HSP47. I would suggest something like this: "HSP47 levels determine the degree of body adiposity".
Introduction: line 84-98. The information about adipose tissue is redundant.

Author's response:

Thank you for the valuable comments. As suggested by the reviewer, we have revised the title to "HSP47 determines the amount of body fat." We have also removed the redundant part of the introduction (line 84-98) in the revised manuscript, and some of this information has been incorporated into the discussion section as follows:

These findings are consistent with numerous basic and clinical reports that loss of fat tissue **redirects energy deposition towards non-adipose organs, particularly increasing the hepatic glycogen and triglyceride contents, and causes glucose intolerance and insulin resistance, and ultimately the development of diabetes³²⁻³⁴**. The study could serve as a scientific foundation that offers a molecular mechanism of proper fat tissue expansion and its association with systemic energy balance.

----- (Page 11; Line 382-388)

We sincerely appreciate the time you've dedicated to providing your thoughtful suggestions, which have significantly enhanced the quality of the manuscript.